# Uncertainty-inspired Structure Hallucination for Protein-protein Interaction Modeling

## Abstract

Modeling protein-protein interactions (PPI) represents a central challenge within the field of biology, and accurately predicting the consequences of mutations in this context is crucial for various applications, such as drug design and protein engineering. Recent advances in deep learning (DL) have shown promise in forecasting the effects of such mutations. However, the effectiveness of these models is hindered by two primary constraints. First and foremost, obtaining the structures of mutant proteins is a persistent challenge, as they are often elusive to acquire. Secondly, interactions take place dynamically, but dynamics is rarely integrated into the DL architecture design. To address these obstacles, we present a novel framework known as Refine-PPI, which incorporates two key enhancements. On the one hand, we introduce a structure refinement module that is trained by a mask mutation modeling (MMM) task on available wide-type structures and then is transferred to hallucinate the inaccessible mutant protein structures. Additionally, we employ a new kind of geometric networks to capture the dynamic 3D variations and encode the uncertainty associated with PPI. Through comprehensive experiments conducted on the established benchmark dataset SKEMPI, our results substantiate the superiority of the Refine-PPI framework. These findings underscore the effectiveness of our hallucination strategy in addressing the absence of mutant protein structure and hope to shed light on the prediction of the free energy change.

## 1 Introduction

Proteins seldom act in isolation and typically engage in interactions with other proteins to perform a wide array of biological functions (Phizicky & Fields, 1995; Du et al., 2016). One illustrative instance involves antibodies, which belong to a category of proteins within the immune system. They identify and attach to proteins found on pathogen surfaces and trigger immune responses by interacting with receptor proteins in immune cells (Lu et al., 2018). Accordingly, it is crucial to devise approaches to modulate these interactions, and a prevalent manipulation strategy is to introduce amino acid mutations at the interface (see Figure 2). However, the space of possible mutations is vast, making it impractical or cost prohibitive to conduct experimental tests on all viable modifications in a laboratory setting (Li et al., 2023). As a consequence, computational techniques are required to guide the recognition of desirable mutations by forecasting their mutational effects on binding strength, which are commonly measured by the change in binding free energy termed $\Delta\Delta G$.

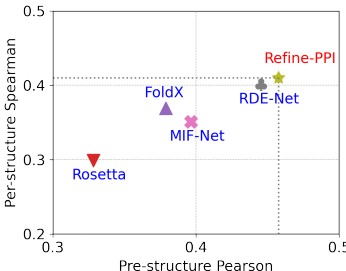

Figure 1: Performance of Refine-PPI on SKEMPI v2 compared to other energy-based or pretrained baselines. Our supervised-only approach achieves the best per-structure correlation metrics.

The past decade has witnessed great potential of deep learning (DL) techniques (Rives et al., 2021; Wu et al., 2022a; Min et al., 2022; Wu et al., 2022c) in modeling proteins. They are employed to a broad range of applications in biological science, such as protein design (Jing et al., 2020), folding

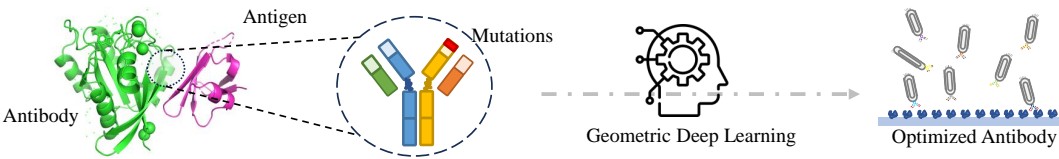

Figure 2: Geometric deep learning is applied to optimize the antibody sequences and achieve desired properties (*e.g.*, better affinity and specificity).

classification (Hermosilla et al., 2020), model quality assessment (Wu et al., 2023), and function prediction (Gligorijević et al., 2021). These DL algorithms also surpass a variety of conventional approaches in computing $\Delta\Delta G$, which can be roughly divided into biophysics-based and statistics-based kinds. In particular, biophysics-based methods depend on sampling from energy functions and consequently face a trade-off between efficiency and accuracy (Schymkowitz et al., 2005; Leman et al., 2020). Meanwhile, statistical-based methods are limited by the selection of descriptors and cannot take advantage of the growing availability of protein structures (Alford et al., 2017).

Despite the fruitful progress made by DL in identifying the free energy change, their efficacy continues to encounter various obstacles. First is the absence of the mutant complex structure. Due to the long-standing consensus that the function of a protein is intricately related to its structure (Jumper et al., 2021), an emerging line of research seeks to encode protein structures using 3D-CNNs or Graph Neural Networks (GNNs) (Jing et al., 2020; Satorras et al., 2021). However, they typically rely on experimental protein structures, specifically those of the Protein Data Bank (PDB), and their performance deteriorates significantly when fed low-quality or noisy protein structures Wu et al. (2022a). Regrettably, in real-world scenarios that involve antibody optimization, obtaining the mutant structure is an insurmountable obstacle, and the exact conformational variations upon mutations are unknown. While groundbreaking approaches such as Alphafold (Jumper et al., 2021) and Alphafold-Multimer (Evans et al., 2021) have brought a revolution in directly inferring a protein's 3D structure from its amino acid sequence, they struggle to accurately forecast the structure of antibody-antigen complexes when compared to monomeric proteins (Ruffolo et al., 2023). As an alternative, some scientists turn to energy-based protein folding tools like FoldX (Delgado et al., 2019) to sample mutant structures, which show finite efficacy and, more importantly, dramatically increase overall computational time (Cai et al., 2023). The second limitation is that the present DL mechanisms often overlook the fundamental thermodynamic principle. It is widely recognized that proteins exhibit inherent dynamism, and these dynamic properties are critical for their biological functions and therapeutic targeting (Miller & Phillips, 2021). Many observations in the real world are not solely dependent on a single molecular structure, but are influenced by the equilibrium distribution of structures (Ganser et al., 2019). For example, inferring biomolecule functions involves assessing the probabilities associated with various structures to identify metastable states. Statistical methods that incorporate probabilistic densities within the structural space enable computation of essential thermodynamic properties, such as entropy and free energies.

To overcome these barriers, we introduce a novel framework named Refine-PPI (see Figure 3) with two key innovations for the mutation effect prediction problem. Firstly, we devise a masked mutation modeling (MMM) strategy and propose to simultaneously predict the mutant structure and $\Delta\Delta G$. Refine-PPI combines the prediction of structure and the prediction of free energy change into a joint training objective rather than relying on external software to sample mutant structures, which offers several distinct advantages. On the one hand, hallucinated mutant structure exhibits significant differences from the wide-type structure, providing crucial geometric information related to the change in binding free energy. On the other hand, MMM not only enables inference of the most likely equilibrium conformation of the mutant structure, but also encourages graph manifold learning with the denoising objective Godwin et al. (2021). Besides, the free energy change implicitly conveys extra information about the structural difference before and after the mutation. Collective training with $\Delta\Delta G$ would definitely promotes the efficiency of structure prediction. Last but not least, in this study, we introduce a new sort of geometric GNN dubbed PDC-Net to capture the flexibility and dynamics of conformations during the binding process. Specifically, each particle in a complex is represented as a probability density cloud (PDC) that illustrates the scale and strength of their motion throughout the interation procedure. Comprehensive evaluation of the SKEMPI

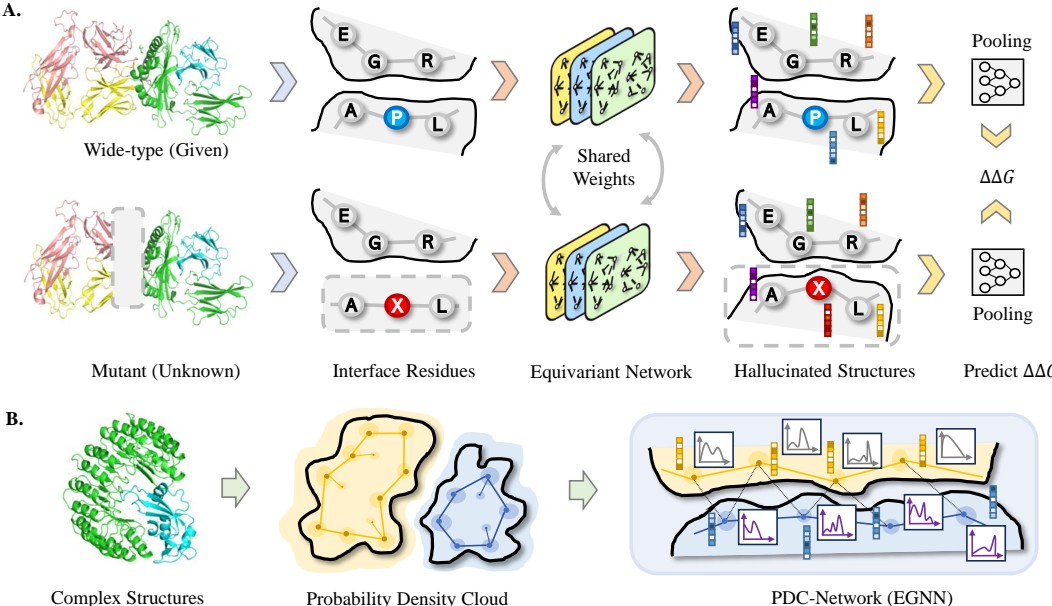

Figure 3: **A.** Refine-PPI pipeline. The given wide-type structure and the masked mutant structure are subsequently fed into weight-shared equivariant graph neural networks (EGNNs). The masked region is reconstructed, and the mutation effect is predicted by comparing features of two resulting complexes. **B.** Deep learning architecture. The particles in the complex are represented as probability density clouds (PDCs) and then encoded by PDC-Net to propagate geometric distributions.

dataset (Jankauskaitė et al., 2019) proves the superiority of our Refine-PPI and demonstrates that it is promising to generate absent mutant structures (see Figure1).

## 2 METHOD AND MATERIALS

### 2.1 PRELIMINARY AND BACKGROUND

**Definition and Notations.** A protein-protein complex is a multi-chain protein structure that can be separated into two groups. Each group contains at least one protein chain and each chain consists of several amino acids. The wide-type complex is usually represented as a 3D geometric graph $\mathcal{G}^{\mathrm{WT}}$, constituted of a ligand $\mathcal{G}_{\mathrm{L}}^{\mathrm{WT}}$ and a receptor $\mathcal{G}_{\mathrm{R}}^{\mathrm{WT}}$. Each graph $\mathcal{G}$ is composed of a batch of nodes $\mathcal{V}$ and edges $\mathcal{E}$. Nodes $\mathcal{V}$ can represent residues or atoms at different resolutions, and every node $v_i \in \mathcal{V}$ has several intrinsic attributes such as the initial $\psi_h$-dimension roto-translational invariant features $\mathbf{h}_i \in \mathbb{R}^{\psi_0}$ (*e.g.*, atom or amino acid types, and electronegativity) and coordinates $\mathbf{x}_i \in \mathbb{R}^3$. The edges $\mathcal{E}$ determine the connectivity between these particles and can be divided into internal edges within each component as $\mathcal{E}_{\mathrm{L}}$ and $\mathcal{E}_{\mathrm{R}}$ and external edges between counterparts as $\mathcal{E}_{\mathrm{LR}}$. Furthermore, we assume that there are $n$ residues in the entire complex and the residue numbers are consistent (*i.e.*, $\left|\mathcal{V}^{\mathrm{WT}}\right| = \left|\mathcal{V}^{\mathrm{MT}}\right| = n$). In our case, we select 4 backbone atoms $\{\mathrm{N}, \mathrm{C}_\alpha, \mathrm{C}, \mathrm{O}\}$ and an additional $\mathrm{C}_\beta$ to represent each amino acid.

**Problem Statement.** The task of predicting the mutation effect can be formulated as approximating the ground-truth function that maps from the wide-type structure $\mathcal{G}^{\mathrm{WT}}$ and mutant information (*i.e.*, where and how some residues mutate from one type $a_i \in \{\mathrm{ACDEFGHIKLMNPQRSTVWY}\}$ to the other $a_i'$) to the change in the binding free energy $\Delta\Delta G$.

### 2.2 STRUCTURAL HALLUCINATION

**Overview of Refine-PPI.** Refine-PPI (see Figure 3) is made up of three major constituents, parameterized by $\rho, \theta, \tau$, respectively. Explicitly, the backbone module $h_\rho(.)$ encodes the input 3D

complex structure, the structure refinement module $f_\theta(.)$ plays the role of hallucinating the unseen mutant structure, and the predictor $g_\tau(.)$ is used for the final $\Delta\Delta G$ estimation. The whole pipeline is described below. To begin with, the wide-type structure $\mathcal{G}^{\text{WT}}$ and a well-initialized mutant structure $\tilde{\mathcal{G}}^{\text{MT}}$ (the initialization details will be elucidated later) are fed into the encoder $h_\rho(.)$ to gain their corresponding features $\mathbf{Z}^{\text{WT}} \in \mathbb{R}^{n\times\psi_1}$ and $\mathbf{Z}^{\tilde{\text{MT}}} \in \mathbb{R}^{n\times\psi_1}$, respectively. Then, the imperfect mutant structure $\tilde{\mathcal{G}}^{\text{MT}}$ along with its first-round representation $\mathbf{Z}^{\tilde{\text{MT}}}$ is forwarded into the structure refinement module $f_\theta(.)$ for several cycles and acquires the ultimate structure $\hat{\mathcal{G}}^{\text{MT}}$ with more robust coordinates $\hat{\mathbf{x}}^{\text{MT}}$. Subsequently, the hallucinated mutant structure $\hat{\mathcal{G}}^{\text{MT}}$ is encoded by $h_\rho(.)$ again, and we can retrieve its second-round updated representation $\mathbf{Z}^{\text{MT}} \in \mathbb{R}^{n\times\psi_1}$. As last, a pooling layer and $g_\tau(.)$ are appended to aggregate graph-level representations of both wide-type and mutation-type noted as $\mathbf{H}^{\text{WT}} \in \mathbb{R}^{\psi_2}$ and $\mathbf{H}^{\text{MT}} \in \mathbb{R}^{\psi_2}$ based on $\mathbf{Z}^{\text{WT}}$ and $\mathbf{Z}^{\text{MT}}$, and output the predicted free energy change $\hat{y}$.

**Mask Mutation Modeling.** Since the ground truth mutant structure $\mathcal{G}^{\text{MT}}$ is hard to attain, we rely on the accessible $\mathcal{G}^{\text{WT}}$ to endow our structure refinement module $f_\theta(.)$ with the capability to restore the fragmentary structures. To this end, we introduce a mask mutation modeling (MMM) task, which requires $f_\theta(.)$ to reconstruct corrupted wide-type structures $\tilde{\mathcal{G}}^{\text{WT}}$. Here, we consider a single-mutation circumstance for better illustration and assume that the $m$-th residue mutates from $a_m$ to $a'_m$. Then, a $(l+r)$-length segment around this mutation site is masked, denoted as $\mathcal{V}_{\text{mut}} = \{v_i\}_{i=m-l}^{m+r}$, which starts from the $(m-l)$-th residue and ends at the $(m+r)$-th residue. Our aim is to recover the structure of this masked region $\{\mathbf{x}^{\text{WT}}\}_{i=m-l}^{m+r}$ given the disturbed complex structure $\tilde{\mathcal{G}}^{\text{WT}}$ and its corresponding representation, as well as the native amino acid type $a_m$. The entire process can be written as follows.

$$f_\theta\left(\mathbf{Z}^{\tilde{\text{MT}}}, \tilde{\mathcal{G}}^{\text{WT}}, a_m\right) \rightarrow \left\{\mathbf{x}^{\text{WT}}\right\}_{i=m-l}^{m+r}. \tag{1}$$

Intuitively, how to corrupt the wide-type structure $\mathcal{G}^{\text{MT}}$ is significant to the success of our MMM, since during the inference time, the same corruption mechanism will be imposed to procure the incipient mutant structure $\tilde{\mathcal{G}}^{\text{MT}}$, which serves as a starting point to deduce the final hallucinated structure $\hat{\mathcal{G}}^{\text{MT}}$. Here, we investigate two kinds of strategy to initialize the coordinates of of entities within the masked regions $\mathcal{V}_{\text{mut}}$. Firstly, we borrow ideas from denoising-based molecular pretraining methods (Godwin et al., 2021; Feng et al., 2023) and independently add a random Gaussian noise of zero mean $\epsilon \sim \mathcal{N}(\mathbf{0}, \boldsymbol{\alpha})$ to the original coordinates as $\tilde{\mathbf{x}}_i^{\text{WT}} = \mathbf{x}_i^{\text{WT}} + \epsilon$, where $\boldsymbol{\alpha}$ determines the scale of the noisy deviation. This denoising objective has been shown to be equivalent to learning a special force field (Zaidi et al., 2022).

In addition to that, we introduce a significantly more challenging mode to corrupt the wide-type structure $\mathcal{G}^{\text{MT}}$ and hypothesize that the mutant regions $\mathcal{V}_{\text{mut}}$ are completely unknown. To be specific, we initialize the coordinates the masked regions $\{\mathbf{x}^{\text{WT}}\}_{i=m-l}^{m+r}$ according to the even distribution between the residue right before the region (namely, $v_{m-l-1}$) and the residue right after the region (namely, $v_{m+r+1}$). Notably, the situation can occur when the residue immediately preceding or following the region does not exist, in which case we extend the existing side in reverse to initialize $\mathcal{V}_{\text{mut}}$ (see Figure 6). The overall initialization process is mathematically written as follows:

$$\tilde{\mathbf{x}}_i = \begin{cases} \mathbf{x}_{m-l-1} + (i-m+l+1)\frac{\mathbf{x}_{m+r+1}-\mathbf{x}_{m-l-1}}{l+r+2}, & \text{if } \exists v_{m-l-1}, v_{m+r+1}, \\ \mathbf{x}_{m+r+1} - (m+r+1-i)\left(\mathbf{x}_{m+r+2}-\mathbf{x}_{m+r+1}\right), & \text{if } \nexists v_{m-l-1}, \exists v_{m+r+1}, \\ \mathbf{x}_{m-l-1} + (i-m+l+1)\left(\mathbf{x}_{m-l-1}-\mathbf{x}_{m-l-2}\right), & \text{if } \exists v_{m-l-1}, \nexists v_{m+r+1}, \end{cases} \tag{2}$$

Noteworthily, both initialization strategies can be easily extended to multiple mutations.

After that, the corrupted wide-type structure $\tilde{\mathcal{G}}^{\text{WT}}$ is sent sequentially to the geometric encoder $h_\rho(.)$ and the structure refinement module $f_\theta(.)$ to restore the coordinates of the mutant regions masked, resulting in $\hat{\mathbf{x}}^{\text{WT}}$. Due to the fact that coordination data usually contains noise, we take the cue from MEAN (Kong et al., 2022) and adopt the Huber loss (Huber, 1992) instead of the common RMSD loss to avoid numerical instability. The loss function is defined as follows by comparing to the actual coordinates $\mathbf{x}_i$:

$$\mathcal{L}_{\text{refine}} = \sum_{i\in\mathcal{V}_{\text{mut}}} \frac{1}{|\mathcal{V}_{\text{mut}}|} l_{\text{huber}}(\hat{\mathbf{x}}_i, \mathbf{x}_i). \tag{3}$$

---

**Algorithm 1** The workflow of our Refine-PPI.

---

**Input:** wide-type structure $\mathcal{G}^{\text{WT}}$, mutant site and amino acid types $a_m$ and $a'_m$; backbone module $h_\rho$, refinement model $f_\theta$, head predictor $g_\tau$; number of recycles $k$, the real free energy change $y$, loss weight $\lambda$

$\tilde{\mathcal{G}}_0^{\text{WT}}, \tilde{\mathcal{G}}_0^{\text{MT}} \leftarrow$ Equation 2 $\left(\mathcal{G}^{\text{WT}}\right)$     $\triangleright$ Initialize wide-type and mutant structures

\# Training-only

**for** $t = 0, 1, ..., k-1$ **do**

    $\mathbf{Z}_t^{\text{WT}} \leftarrow h_\rho\left(\tilde{\mathcal{G}}_t^{\text{WT}}\right)$

    $\tilde{\mathbf{x}}_{t+1}^{\text{WT}} \leftarrow f_\theta\left(\tilde{\mathcal{G}}_t^{\text{WT}}, \mathbf{Z}_t^{\text{WT}}, \tilde{\mathbf{x}}_t^{\text{WT}}, a_m\right)$

**end for**

$\mathcal{L}_{\text{refine}} \leftarrow$ Equation 3 $\left(\tilde{\mathbf{x}}_k^{\text{WT}}, \mathbf{x}^{\text{WT}}\right)$     $\triangleright$ The MMM loss

**for** $t = 0, 1, ..., k-1$ **do**

    $\mathbf{Z}_t^{\text{MT}} \xleftarrow{\text{No grad.}} h_\rho\left(\tilde{\mathcal{G}}_t^{\text{MT}}\right)$

    $\tilde{\mathbf{x}}_{t+1}^{\text{MT}} \xleftarrow{\text{No grad.}} f_\theta\left(\tilde{\mathcal{G}}_t^{\text{MT}}, \mathbf{Z}_t^{\text{MT}}, \tilde{\mathbf{x}}_t^{\text{MT}}, a'_m\right)$

**end for**

$\mathbf{Z}^{\text{WT}}, \mathbf{Z}^{\text{MT}} \leftarrow h_\rho\left(\mathcal{G}^{\text{WT}}\right), h_\rho\left(\tilde{\mathcal{G}}_k^{\text{MT}}\right)$

$\hat{y} \leftarrow g_\tau\left(\mathbf{Z}^{\text{WT}}, \mathbf{Z}^{\text{MT}}\right)$

$\mathcal{L}_{\Delta\Delta G} \leftarrow \text{RMSE}(\hat{y}, y)$     $\triangleright$ The $\Delta\Delta G$ loss

\# Backpropagation

$\rho, \theta, \tau \leftarrow \mathcal{L}_{\Delta\Delta G} + \lambda\mathcal{L}_{\text{refine}}$

---

$\Delta\Delta G$ **Prediction.** We impose the same strategy in MMM to initialize the mutant structure, gaining $\tilde{\mathcal{G}}^{\text{MT}}$ based on $\mathcal{G}^{\text{WT}}$. Then given the mutant information $a'_m$, we utilize the weight-shared encoder $h_\rho(.)$ and the weight-shared structure refinement module $f_\theta(.)$ to hallucinate the unknown mutant structure as $p\left(\left\{\mathbf{x}^{\text{MT}}\right\}_{i=m-l}^{m+r} \Big| \tilde{\mathcal{G}}^{\text{MT}}, a'_m, \theta, \rho\right)$. It is worth noting that the resulting $\hat{\mathbf{x}}^{\text{WT}}$ does not carry gradients and we do not expect to perform backpropagation at this phase. Later, we leverage the original wide-type structure $\mathcal{G}^{\text{WT}}$ and the refined mutant structure $\hat{\mathcal{G}}^{\text{MT}}$ to extract their corresponding representations $\mathbf{Z}^{\text{WT}}$ and $\mathbf{Z}^{\text{MT}}$, separately. $\mathbf{Z}^{\text{WT}}$ and $\mathbf{Z}^{\text{MT}}$ are then delivered to the regressor $g_\tau(.)$ to acquire the change in free energy $\hat{y}$. Total supervision is realized by the sum of two losses as $\mathcal{L} = \mathcal{L}_{\Delta\Delta G}(y, \hat{y}) + \lambda\mathcal{L}_{\text{refine}}\left(\left\{\mathbf{x}^{\text{WT}}\right\}_{i=m-l}^{m+r}, \left\{\hat{\mathbf{x}}^{\text{WT}}\right\}_{i=m-l}^{m+r}\right)$, where $\lambda$ is the balance hyperparameter. The whole training paradigm is illustrated in pseudo-code 1.

**Discussion.** Previous studies exemplified by Google's DeepDream (Mordvintsev et al., 2015) train networks to recognize faces and other patterns in images, and invert and adjust arbitrary input images to draw more strongly resemble faces or other patterns perceived by the network. The generated images are often referred to as hallucinations because they may not faithfully represent any actual face, but what DL models view as an ideal face. Remarkably, this mechanism has also demonstrated success in the context of macromolecules. It has been shown that information stored in the many parameters of trained networks can be harnessed to design new protein structures featuring new sequences (Anishchenko et al., 2021). In our Refine-PPI, we use a similar methodology and explore whether networks trained on existing wide-type structures could be inverted to generate brand new 'ideal' protein structures based on mutant information. We discover that networks do have the strong hallucination capability to resolve the inevitable dilemma of the missing mutant structures.

## 2.3   PROBABILITY DENSITY CLOUD NETWORK

**Kinetics in Molecules.** In order to fully unleash the potential of our Refine-PPI pipeline, it is crucial to devise an effective geometric network to comprehend protein structures and perform structure refinement. Over the past few years, there has been a surge in the development of cutting-edge architectures aimed at extending networks to work in both Euclidean and non-Euclidean domains, encompassing structures like manifolds, meshes, or strings. Given that molecules can be naturally

represented as graphs, graph-based approaches have become increasingly dominant in molecular modeling (Thomas et al., 2018; Schütt et al., 2018; Fuchs et al., 2020; Liao & Smidt, 2022). Beyond addressing the inherent limitations of GNNs, such as over-smoothing, over-squashing, and representation bottleneck (Wu et al., 2022b), these methods are all dedicated to incorporating geometric principles. Symmetry is a crucial concept in this regard, often expressed through the notions of equivariance and invariance, which describe how systems respond to various transformations. However, previous geometric approaches in molecular science were primarily designed for static and stable molecules characterized by deterministic and uncertainty-free structures. Here, we propose a new technique that takes dynamics into account and integrates it into geometric GNNs.

**Probability Density Cloud.** It is a fundamental concept in physics and chemistry that atoms and molecules are never at rest, even at extremely low temperatures (Clerk-Maxwell, 1873). They exhibit various types of motion, including translational motion (movement from one place to another), rotational motion (spinning or rotating), and vibrational motion (oscillating back and forth). This motivates us to consider microscopic particles in the universe from a kinetic and vibrate perspective rather than an immobile view. Recall that in quantum mechanisms, electrons do not follow well-defined paths like planets around the Sun in classical physics. Instead, they exist at specific energy levels and are described by wave functions, which are mathematical functions that provide information on the probability of finding an electron in various locations around the nucleus (Schumaker, 1986). Physists commonly envision and represent an electron or other quantum particle by depicting the probability distribution of finding them around a specific region of space within an atom or molecule, where the shape and size of these orbitals vary depending on the quantum numbers associated with the electron.

Inspired by this phenomenon, we portray particles as a probability density cloud (PDC) that shows regions in space where there is a higher probability of finding them. For this purpose, we assume that the coordinates of each particle $\mathbf{x}_i$ follow the Gaussian distribution as $\mathcal{N}(\boldsymbol{\mu}_i, \boldsymbol{\Sigma}_i)$. $\boldsymbol{\mu}_i \in \mathbb{R}^3$ is the place where node $i$ is most likely to be located, and $\boldsymbol{\Sigma}_i \in \mathbb{R}^{3 \times 3}$ is a isotropic (or spherical) covariance matrix signifying the independence upon the coordinate system. Given this premise, we can derive a range of invariant geometric characteristics that emphasize molecular structural information. The primary and most crucial variable is the distance, denoted as $d_{ij} = ||\mathbf{x}_i - \mathbf{x}_j||^2$. As $\mathbf{x}_i$ and $\mathbf{x}_j$ are are statistically independent, their difference follows a normal distribution, specifically $\mathbf{x}_i - \mathbf{x}_j \sim \mathcal{N}(\boldsymbol{\mu}_i - \boldsymbol{\mu}_j, \boldsymbol{\Sigma}_i + \boldsymbol{\Sigma}_j)$ (Lemons, 2003). Consequently, the squared norm of this difference, denoted as $d_{ij}^2$, exhibits a generalized chi-squared distribution $\chi^2(.)$ with a set of natural parameters, comprising $(\boldsymbol{\mu}_i - \boldsymbol{\mu}_j, \boldsymbol{\Sigma}_i + \boldsymbol{\Sigma}_j)$. Hence, the mean and variance of this generalized chi-square distribution $\chi^2(.)$, denoted as $\mu_{d_{ij}}$ and $\sigma_{d_{ij}}$, are as follows:

$$\mu_{d_{ij}} = \operatorname{tr}(\boldsymbol{\Sigma}_i + \boldsymbol{\Sigma}_j) + ||\boldsymbol{\mu}_i - \boldsymbol{\mu}_j||^2, \quad \sigma_{d_{ij}} = 2\operatorname{tr}(\boldsymbol{\Sigma}_i + \boldsymbol{\Sigma}_j) + 4(\boldsymbol{\mu}_i - \boldsymbol{\mu}_j)^\top (\boldsymbol{\Sigma}_i + \boldsymbol{\Sigma}_j)(\boldsymbol{\mu}_i - \boldsymbol{\mu}_j),$$
$$(4)$$

where $\operatorname{tr}(.)$ calculates the trace of a matrix. Furthermore, we can also mathematically induce the distributions of some other geometric vaiables. Let $\mathbf{x}_{ab}$ be the directed vector from $\mathbf{x}_a$ to $\mathbf{x}_b$. For example, when considering triangle nodes $(i, j, k)$, the angle distribution $\angle \mathbf{x}_{ij}\mathbf{x}_{ik}$ can be characterized as the distribution of $\arccos \frac{(\mathbf{x}_i - \mathbf{x}_j) \cdot (\mathbf{x}_j - \mathbf{x}_k)}{|\mathbf{x}_i - \mathbf{x}_j||\mathbf{x}_j - \mathbf{x}_k|}$. After establishing the precise first and second moments of distributions of important geometric features, we can now dive into the process of incorporating this dynamic information into geometric GNNs.

**PDC-Net.** Our idea of PDC can be generalized to the majority of existing geometric architectures and here we select equivariant GNN (EGNN) (Satorras et al., 2021) for example, which foregoes computationally intensive high-order representations in intermediate layers while still achieving competitive performance in modeling dynamical systems. The key difference is that PDC-Net no longer accepts geometric deterministic values $d_{ij}$ and $\mathbf{x}_i$, but takes distributions $f_{d_{ij}}$ and $f_{\mathbf{x}_i}$ as ingredients. Its $l$ layer, named PDC-L, takes the set of node embeddings $\mathbf{h}^{(l)} = \left\{\mathbf{h}_i^{(l)}\right\}_{i=1}^n$, edge information $\mathcal{E} = \{\mathcal{E}_L, \mathcal{E}_R, \mathcal{E}_{LR}\}$, and geometric feature distributions $\boldsymbol{\nu}^{(l)} = \left\{\boldsymbol{\mu}_i^{(l)}, \boldsymbol{\Sigma}_i^{(l)}\right\}_{i=1}^n$ as input, and outputs a transformation on $\mathbf{h}^{(l+1)}$ and $\boldsymbol{\nu}^{(l+1)}$. Concisely,

$\mathbf{h}^{(l+1)}, \boldsymbol{\nu}^{(l+1)} = \text{PDC-L} \left[ \mathbf{h}^{(l)}, \boldsymbol{\nu}^{(l)}, \mathcal{E} \right]$, which is defined as follows:

$$\mathbf{m}_{j \rightarrow i} = \phi_e \left( \mathbf{h}_i^{(l)}, \mathbf{h}_j^{(l)}, \mu_{d_{ij}}^{(l)}, \sigma_{d_{ij}}^{(l)} \right), \quad \mathbf{h}_i^{(l+1)} = \phi_h \left( \mathbf{h}_i^{(l)}, \sum_j \mathbf{m}_{j \rightarrow i}, \right), \tag{5}$$

$$\boldsymbol{\mu}_i^{(l+1)} = \boldsymbol{\mu}_i^{(l)} + \frac{1}{|\mathcal{N}(i)|} \sum_{j \in \mathcal{N}(i)} \left( \boldsymbol{\mu}_i^{(l)} - \boldsymbol{\mu}_j^{(l)} \right) \phi_\mu(\mathbf{m}_{j \rightarrow i}), \tag{6}$$

$$\boldsymbol{\Sigma}_i^{(l+1)} = \boldsymbol{\Sigma}_i^{(l)} + \frac{1}{|\mathcal{N}(i)|} \sum_{j \in \mathcal{N}(i)} \left( \boldsymbol{\Sigma}_i^{(l)} + \boldsymbol{\Sigma}_j^{(l)} \right) \phi_\sigma(\mathbf{m}_{j \rightarrow i}), \tag{7}$$

where $\phi_e, \phi_h, \phi_\mu, \phi_\sigma$ are the edge, node, mean, and variance operations respectively that are commonly approximated by Multilayer Perceptrons (MLPs). It is worth noting that the mean position of each particle, denoted as $\boldsymbol{\mu}_i$, is updated through a weighted sum of all relative differences $(\boldsymbol{\mu}_i - \boldsymbol{\mu}_j)_{\forall j \in \mathcal{N}(i)}$. Meanwhile, the variance $\boldsymbol{\Sigma}_i$ is updated by a weighted sum of all additions $(\boldsymbol{\Sigma}_i + \boldsymbol{\Sigma}_j)_{\forall j \in \mathcal{N}(i)}$. These strategies align with the calculation of the mean and variance of the difference between two normal random variables. We also provide another type of mechanism to update the variance and observe a slight improvement in Appendix B.3. As for the initialization of coordinate variance $\boldsymbol{\Sigma}$, we explore three sorts of different approaches and details are elucidated in Appendix B.2. Moreover, it is readily apparent that PDC-Net maintains the equivariance property, and the proof can be found in the Supplementary Note D.

## 3 EXPERIMENTS

### 3.1 EXPERIMENTAL SETUPS

**Data and Metrics.** Evaluation is carried out in the widely recognized SKEMPI.v2 database (Jankauskaitė et al., 2019). It contains data on changes in the thermodynamic parameters and kinetic rate constants after mutation for structurally resolved protein–protein interactions. The latest version contains manually curated binding data for 7,085 mutations. The dataset is splitted into 3 folds by structure, each containing unique protein complexes that do not appear in other folds. Two folds are used for train and validation, and the remaining fold is used for test. This yields 3 different sets of parameters and ensures that every data point in SKEMPI.v2 is tested once.

Similarly to Luo et al. (2023), we use five metrics to evaluate the accuracy of $\Delta\Delta G$ predictions, including Pearson and Spearman correlation coefficients, minimized RMSE, minimized MAE (mean absolute error) and AUROC (area under the receiver operating characteristic). Calculating AUROC involves classifying mutations according to the direction of their $\Delta\Delta G$ values. In practical scenarios, the correlation observed within a specific protein complex attracts heightened interest. To account for this, we arrange mutations according to their associated structures. Groups with fewer than 10 mutation data points are excluded from this analysis. Subsequently, correlation calculations are performed for each structure independently. This introduces two additional metrics: **the average per-structure Pearson and Spearman correlation coefficients**. Other experimental details are explained in the Appendix A.

**Baselines.** We evaluate the effectiveness of our PDC-Net against various categories of techniques. The initial kind encompasses conventional empirical energy functions such as **Rossetta** Cartesian $\Delta\Delta G$ Park et al. (2016); Alford et al. (2017) and **FoldX**. The second grouping comprises sequence/evolution-based methodologies, exemplified by **ESM-1v** Meier et al. (2021), **PSSM** (position-specific scoring matrix), **MSA Transformer** Rao et al. (2021), and Tranception Notin et al. (2022). The third category includes end-to-end learning models such as **DDGPred** Shan et al. (2022) and another **End-to-End** model that adopts Graph Transformer (GT) Luo et al. (2023) as the encoder architecture, but employs an MLP to directly forecast $\Delta\Delta G$. The fourth grouping encompasses unsupervised/semi-supervised learning approaches, consisting of **ESM-IF** Hsu et al. (2022) and Masked Inverse Folding (MIF) Yang et al. (2022). They first pretrain networks on structural data and then employ the pretrained representations to predict $\Delta\Delta G$. MIF also utilizes GT as an encoder for comparative purposes with two variations: **MIF-$\Delta$logit** uses the disparity in log-probabilities of

Table 1: Evaluation of $\Delta\Delta G$ prediction on the SKEMPI.v2 dataset.

| Method | Pretrain | Per-Structure | | Overall | | | | |
| --- | --- | --- | --- | --- | --- | --- | --- | --- |
| | | Pearson | Spearman | Pearson | Spearman | RMSE | MAE | AUROC |
| **Energy Function-based** | | | | | | | | |
| Rosetta | – | 0.3284 | 0.2988 | 0.3113 | 0.3468 | 1.6173 | 1.1311 | 0.6562 |
| FoldX | – | 0.3789 | 0.3693 | 0.3120 | 0.4071 | 1.9080 | 1.3089 | 0.6582 |
| **Supervised-based** | | | | | | | | |
| DDGPred | ✗ | 0.3750 | 0.3407 | 0.6580 | 0.4687 | **1.4998** | **1.0821** | 0.6992 |
| End-to-End | ✗ | 0.3873 | 0.3587 | 0.6373 | 0.4882 | 1.6198 | 1.1761 | 0.7172 |
| **Sequence-based** | | | | | | | | |
| ESM-1v | ✓ | 0.0073 | -0.0118 | 0.1921 | 0.1572 | 1.9609 | 1.3683 | 0.5414 |
| PSSM | ✓ | 0.0826 | 0.0822 | 0.0159 | 0.0666 | 1.9978 | 1.3895 | 0.5260 |
| MSA Transf. | ✓ | 0.1031 | 0.0868 | 0.1173 | 0.1313 | 1.9835 | 1.3816 | 0.5768 |
| Tranception | ✓ | 0.1348 | 0.1236 | 0.1141 | 0.1402 | 2.0382 | 1.3883 | 0.5885 |
| **Unsupervised or Semi-supervised-based** | | | | | | | | |
| B-factor | ✓ | 0.2042 | 0.1686 | 0.2390 | 0.2625 | 2.0411 | 1.4402 | 0.6044 |
| ESM-IF | ✓ | 0.2241 | 0.2019 | 0.3194 | 0.2806 | 1.8860 | 1.2857 | 0.5899 |
| MIF-$\Delta$logit | ✓ | 0.1585 | 0.1166 | 0.2918 | 0.2192 | 1.9092 | 1.3301 | 0.5749 |
| MIF-Net. | ✓ | 0.3965 | 0.3509 | 0.6523 | 0.5134 | 1.5932 | 1.1469 | 0.7329 |
| RDE-Linear | ✓ | 0.2903 | 0.2632 | 0.4185 | 0.3514 | 1.7832 | 1.2159 | 0.6059 |
| RDE-Net. | ✓ | 0.4448 | 0.4010 | 0.6447 | 0.5584 | 1.5799 | 1.1123 | 0.7454 |
| Refine-PPI | ✗ | 0.4475 | 0.4102 | 0.6584 | 0.5394 | 1.5556 | 1.0946 | 0.7517 |
| Refine-PPI | ✓ | **0.4561** | **0.4374** | **0.6592** | **0.5608** | 1.5643 | 1.1093 | **0.7542** |

Figure 4: **A.** Visualization of correlations between experimental $\Delta\Delta G$ and predicted $\Delta\Delta G$. **D.** A selected exapmle of a predicted mutant structure's interface. **B.** The scatter plot shows that the error of wide-type structure recovery has a positive relation with the error of $\Delta\Delta G$ prediction.

amino acid types to attain $\Delta\Delta G$, and **MIF-Network** predicts $\Delta\Delta G$ based on the acquired representations. Besides, **B-factors** is the network that anticipates the B-factor of residues and incorporate the projected B-factor in lieu of entropy for $\Delta\Delta G$ prediction. Lastly, Rotamer Density Estimator (RDE) Luo et al. (2023) uses a flow-based generative model to estimate the probability distribution of rotamers and uses entropy to measure flexibility with two variants containing **RDE-Linear** and **RDE-Network**. More details on the implementation can be found in Appendix A.

## 3.2 RESULTS

**Comparison with Baselines.** Table 1 documents the results, and performance on subsets of single-mutation and multi-mutation are removed to Appendices 4 and 5 due to the space limitation. Our Refine-PPI model is competitive and better in all regression metrics. Precisely, it achieves the highest per-structure Spearman and Pearson's correlations, which are considered as our primary metrics because the correlation of one specific protein complex is the most important.

In particular, multiple point mutations have been shown to be often required for successful affinity maturation (Sulea et al., 2018), and Refine-PPI outperforms DDGPred and RDE-Net by a large margin in the multi-mutation subset. This stems from the fact that RDE-Net and DDGPred perceive the mutant structures the same as the wide-type structures and consequently are not aware of the structural distinction. On the contrary, the mutant structures with multiple mutations should be more different than those with single mutations, and therefore it becomes more crucial to detect the

Table 2: Ablation study of Refine-PPI without pretraining, where we choose the backbone $h_\rho$ (*i.e.*, Graph Transformer) as the foundation model for comparison (*i.e.*, No. 1).

| No. | MMM | PDC-Net | Per-Structure | | Overall | | | | |
| --- | --- | --- | --- | --- | --- | --- | --- | --- | --- |
| | | | Pearson | Spearman | Pearson | Spearman | RMSE | MAE | AUROC |
| 1 | ✗ | ✗ | 0.3708 | 0.3353 | 0.6210 | 0.4907 | 1.6199 | 1.1933 | 0.7225 |
| 2 | ✓ | ✗ | 0.4145 | 0.3875 | 0.6571 | 0.5553 | 1.5580 | 1.1025 | 0.7460 |
| 3 | ✓ | ✓ | **0.4475** | **0.4102** | **0.6584** | **0.5394** | **1.5556** | **1.0946** | **0.7517** |

variant after the mutation. Refine-PPI anticipates the structural transformation due to mutation and is capable of connecting the structural change with $\Delta\Delta G$. Notably, strong baselines such as RDE-Net, MIF-Net, ESM-IF enjoy the benefits of unsupervised pretraining on PDB-REDO. Meanwhile, Refine-PPI trained from scratch has already outpassed these pretrained methodologies. This further verifies the great success of our Refine-PPI framework.

**Visualization.** We visualize three hallucinated mutant structures in Appendix C. In addition, we envision the scatter plot of experimental and predicted $\Delta\Delta G$ and also draw the relation between the error of wide-type structure recovery and the error of $\Delta\Delta G$ estimation in Figure 4. It can be found that, generally, a small error of wide-type structure reconstruction leads to a more accurate $\Delta\Delta G$ prediction. This indicates that these two tasks are closely related to each other. In addition, we randomly pick up four exemplary PDBs and visualize the learned variance of our PDC-Net, that is, the magnitude of $||\Sigma_i||^2$ in Figure 5 and quantitative analysis in Appendix B.4. Pictures show that particles at the interface have a smaller variation compared to those at the edges of proteins. This aligns with the biological concept that atoms in the binding surface are less volatile than atoms in other parts of the complex. This phenonmenon confirms that PDC-Net has adaptively comprehended the magnitude and strength of entities' motion during PPI. Finally, we also provide a case study of 16 seed complexes with different numbers of mutations that are well predicted by our Refine-PPI in the Appendix 7.

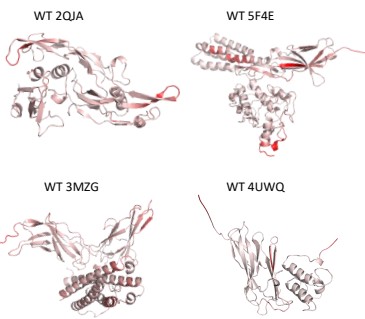

Figure 5: Visualization of the thermodyanmics learned by PDC-Net within several complexes, where a darker color corresponds to a more flexible protein segment.

**Ablation Studies.** We also conduct additional experiments to investigate the contributions of each components of our Refine-PPI and the results are displayed in Table 2. It can be concluded that the introduction of co-training of the structure refinement and the $\Delta\Delta G$ prediction greatly contributes to the promotion of all metrics, cultimating in an increase of $11.8\%$ and $15.6\%$ in per-structure Pearson's and Spearman correlations. Additionally, PDC-Net also brings obvious benefits such as a lower MAE and a higher AUORC.

Table 3: Performance of different coordinate initialization strategies for MMM.

| Method | Per-Structure | |
| --- | --- | --- |
| | Pearson | Spearman |
| Easy | 0.4417 | 0.4060 |
| Hard | **0.4475** | **0.4102** |

In Table 3, we report the performance of two initialization strategies to corrupt the masked region. The easy mode (denoising-based) is slightly outpassed by the hard one (surroundings-based).

## 4 CONCLUSION

In this work, we propose a new framework named Refine-PPI to predict the mutation effect. Given the circumstance that mutant structures are always absent, we introduce an additional structure refinement module to recover the masked regions around the mutations. This module is trained simultaneously via mask geometric modeling. In addition to that, we notice that protein-protein interactions are a dynamic process, but few prior studies have taken this characteristic into account in a deep learning design. To bridge the gap, we present a probablity density cloud (PDC)-Network to capture the dynamics in atomic resolution. Our results highlight the necessity to adopt a more robust mutant structure and consider dynamics for molecular modeling. A statement regarding limitation and future work is elaborated in Appendix E.

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

# A EXPERIMENTAL DETAILS

We implement all experiments on 4 A100 GPUs, each with 80G memory. Refine-PPI is trained with an Adam optimizer without weight decay and with $\beta_1 = 0.9$ and $\beta_2 = 0.999$. A ReduceLROn-Plateau scheduler is employed to automatically adjust the learning rate with a patience of 10 epochs and a minimum learning rate of $1.e - 6$. The batch size of is set to 64 and an initial learning rate of $1.e - 4$. The maximum iterations are 50K and the validation frequency is 1K iterations. The node dimension is 128, and no dropout is conducted. For the implementation of all baselines, please refer to Luo et al. (2023) for more details and we directly copy the results from this paper. As for the structure refinement, the number of recycle is set as 3, and the balance weight is tuned as 1.0. We perform a grid search to find the optimal length of the masked region and find that $l = r = 5$ is a good choice. However, different initializations require different optimal hyperparameters, and typically we can mask longer regions for denoising-based MMM.

As for the specific model architecture, the backbone module $h_\rho(.)$ can take the form of any conventional geometric neural networks (e.g., GVP-GNN, EGNN, SE(3)-Transformer, Graph Transformer). Here, we adopt a one-layer Graph Transformer (Luo et al., 2023) to extract general representations of proteins. The refinement module $f_\theta(.)$ needs to output both updated features and coordinates, and therefore we use PDC-EGNN as $f_\theta(.)$ in our experiments. Lastly, the head predictor $g_\tau(.)$ is a simple linear layer that accepts the concatenation of representations of both wide and mutation types and forecasts the change in free energy. The total model size of our Refine-PPI is approximately 6M.

## A.1 BASELINES IMPLEMENTATIONS

Baselines that require training and calibration using the SKEMPI.v2 dataset (DDGPred, End-to-End, B-factor, MIF-$\Delta$logit, MIF-Network, RDE-Linear, and RDE-Net) are trained independently using the 3 different splits of the dataset as described in Section 3.1. This is to ensure that every data point in the SKEMPI.v2 dataset is tested on once. Below are descriptions of the implementation of the baseline methods, which follow the same scheme as Luo et al. (2023)).

**Rosetta** (Alford et al., 2017; Leman et al., 2020): The version we used is 2021.16, and the scoring function is ref2015_cart. Every protein structure in the SKEMPI.v2 dataset is first preprocessed using the relax application. The mutant structure is built by cartesian_ddg. The binding free energies of both wild-type and mutant structures are predicted by interface_energy (dG_separated/dSASAx100). Finally, the binding $\Delta\Delta G$ is calculated by subtracting the binding energy of the wild-type structure from the binding energy of the mutant.

**FoldX** (Delgado et al., 2019): Structures are first relaxed by the RepairPDB command. Mutant structures are built with the BuildModel command based on the repaired structure. The change in binding free energy $\Delta\Delta G$ is calculated by subtracting the wild-type energy from the mutant energy.

**ESM-1v** (Meier et al., 2021): We use the implementation provided in the ESM open-source repository. Protein language models can only predict the effect of mutations for single protein sequences. Therefore, we ignore the cases where mutations occur in multiple sequences. We extract the sequence of the mutated protein chain from the SEQRES entry of the PDB file. We use the masked-marginal mode to score both wild-type and mutant sequences and use their difference as an estimate of $\Delta\Delta G$.

**PSSM** We construct MSAs from the Uniref90 database for chains with mutation annotations in the SKEMPI dataset. We use Jackhmmer version 3.3.1 following the setting in Meier et al. (2021). The MSAs are filtered using HHfilter with coverage 75 and sequence identity 90 . This HHfilter parameter is reported to have the best performance for MSA Transformer according to Meier et al. (2021). We calculate position-specific scoring matrices (PSSM) and use the change in probability as a prediction of $\Delta\Delta G$.

**MSA Transformer** (Rao et al., 2021): We use the implementation provided in the ESM open-source repository. We input the MSAs constructed during the evaluation of the PSSM to the MSA Transformer. We used masked-marginals mode to score both wild-type and mutant sequences and use their difference as the prediction of $\Delta\Delta G$.

**Tranception** (Notin et al., 2022): We use the implementation provided in the Tranception open-source repository. We predict mutation effects using the large model checkpoint. Previously built MSAs (not filtered by HHfilter) are used for inference-time retrieval.

**DDGPred** (Shan et al., 2022): We use the implementation that follows the paper by Shan et al. (2022). Since this model requires predicted sidechain structures of the mutant, we use mutant structures packed during our evaluation of Rosetta to train the model and run prediction.

**End-to-End**: The end-to-end model shares the same encoder architecture as RDE (Luo et al., 2023). The difference is that in the RDE normalizing flows follow the encoder to model rotamer distributions, but in the end-to-end model, the embeddings are directly fed to an MLP to predict $\Delta\Delta G$.

**B-factor**: This model predicts per-atom b-factors for proteins. It has the same encoder architecture as RDE (Luo et al., 2023). The encoder is followed by an MLP that predicts a vector for each amino acid, where each dimension is the predicted b-factor of different atoms in the amino acid. The amino acid-level b-factor is calculated by averaging the atom-level b-factors. The predicted b-factors are used as a measurement of conformational flexibility. They are used to predict $\Delta\Delta G$ using the linear model same as RDE-Linear (Luo et al., 2023).

**ESM-IF** (Hsu et al., 2022): ESM-IF can score protein sequences using the log-likelihood. Implementation of the scoring function is provided in the ESM repository. We enable the –multichain_backbone flag to let the model see the whole protein-protein complex. We subtract the log-likelihood of the wild-type from the mutant to predict $\Delta\Delta G$.

**MIF Architecture**. The masked inverse folding (MIF) network uses the same encoder architecture as RDE (Luo et al., 2023). Following the encoder is a per-amino-acid 20-category classifier that predicts the type of masked amino acids. We use the same PDB-REDO train-test split to train the model. At training time, we randomly crop a patch consisting of 128 residues and randomly mask 10% amino acids. The model learns to recover the type of masked amino acids with the standard cross entropy loss.

**MIF-$\Delta$logit** To score mutations, we first mask the type of mutated amino acids. Then, we use the log probability of the amino acid type as the score. Analogously, we have the score of the wild-type bound ligand, wild-type bound receptor, wild-type unbound ligand, unbound receptor, mutated bound ligand, mutated bound receptor, and mutated unbound ligand. Therefore, we use the identical linear model to RDE-Linear (Luo et al., 2023) to predict $\Delta\Delta G$ from the scores.

**MIF-Network** This is similar to RDE-Network (Luo et al., 2023). The difference is that we use the pre-trained encoder of MIF rather than the encoder of RDE. We also freeze the MIF encoder as we aim to utilize the unsupervised representations.

## A.2   VISUALIZATION OF COORDINATE INITIALIZATION IN MMM

To better clarify the initialization of our MMM, we show the process of two different mechanisms (*i.e.*, the easy denoising-based one and the hard surrounding-based one) in Figure 6.

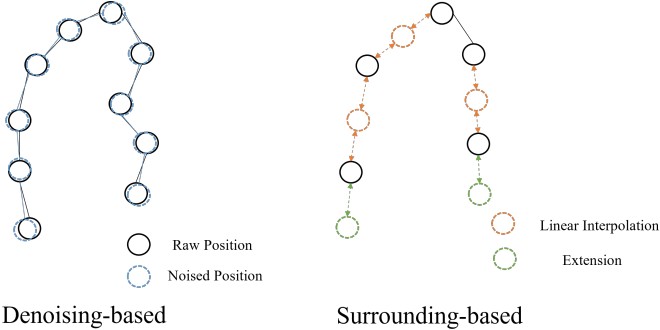

Figure 6: The illustration of coordinate initialization in the MMM task.

# B   ADDITIONAL RESULTS

## B.1   PERFORMANCE ON SUBSETS AND CASE STUDIES

For better comparison of our Refine-PPI and other baselines, we make a bar plot on per-structure Pearson's and Spearman correlations in Figure 8. We also explicitly document the evaluation results of different methods on the multi-mutation and single-mutation subsets of the SKEMPI.v2 dataset in Table 4 and Table 5. It can be found that with pretraining on PDB-REDO, Refine-PPI achieves the best per-structure metrics on both multi-mutation and single-mutation subsets. This indicates that Refine-PPI is a more effective tool to screen and select mutant proteins for desired properties. Moreover, we envision some case studies in SKEMPI.v2 in Figure 7, where we select four examples for each different number of mutations.

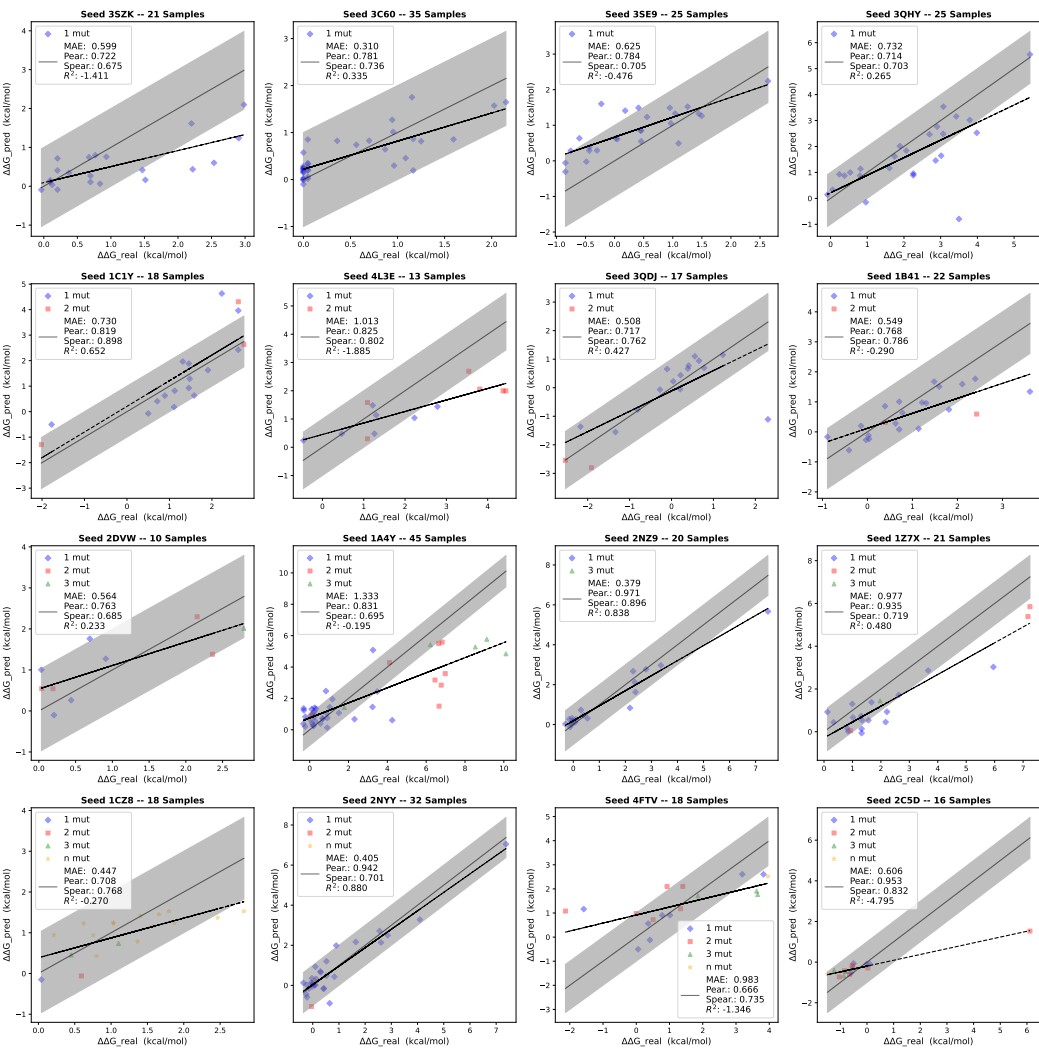

Figure 7: Prediction plots of 16 seed PDBs that are made by Refine-PPI. Four rows correspond to different numbers of mutations, where the grey belt represents acceptable prediction errors. It can be found that Refine-PPI can perform well in all circumstances containing one, two, or more mutations.

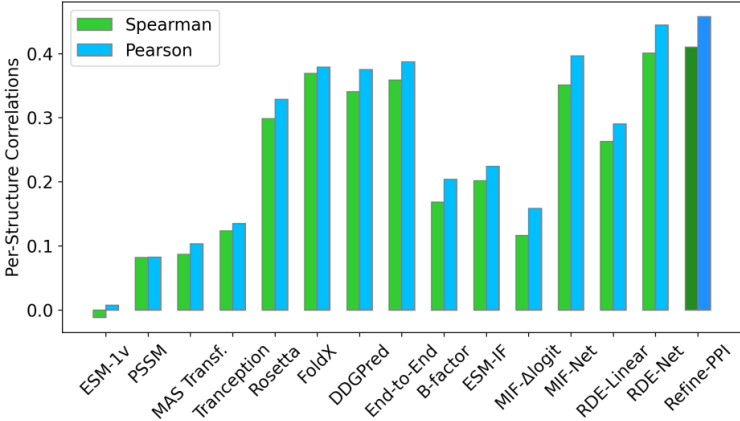

Figure 8: Per-structure Spearman and Pearson correlations of different baseline methods and Refine-PPI.

Table 4: Evaluation of $\Delta\Delta G$ prediction on the multi-mutation subset of the SKEMPI.v2 dataset.

| Method | Pretrain | Per-Structure | | Overall | | | | |
| --- | --- | --- | --- | --- | --- | --- | --- | --- |
| | | Pearson | Spearman | Pearson | Spearman | RMSE | MAE | AUROC |
| **Energy Function-based** | | | | | | | | |
| Rosetta | – | 0.1915 | 0.0836 | 0.1991 | 0.2303 | 2.6581 | 2.0246 | 0.6207 |
| FoldX | – | 0.3908 | 0.3640 | 0.3560 | 0.3511 | 1.5576 | 1.0713 | 0.6478 |
| **Supervised-based** | | | | | | | | |
| DDGPred | ✗ | 0.3912 | 0.3896 | 0.5938 | 0.5150 | 2.1813 | 1.6699 | 0.7590 |
| End-to-End | ✗ | 0.4178 | 0.4034 | 0.5858 | 0.4942 | 2.1971 | 1.7087 | 0.7532 |
| **Sequence-based** | | | | | | | | |
| ESM-1v | ✓ | -0.0599 | -0.1284 | 0.1923 | 0.1749 | 2.7586 | 2.1193 | 0.5415 |
| PSSM | ✓ | -0.0174 | -0.0504 | -0.1126 | -0.0458 | 2.7937 | 2.1499 | 0.4442 |
| MSA Transf. | ✓ | -0.0097 | -0.0400 | 0.0067 | 0.0030 | 2.8115 | 2.1591 | 0.4870 |
| Tranception | ✓ | -0.0688 | -0.0120 | -0.0185 | -0.0184 | 2.9280 | 2.2359 | 0.4874 |
| **Unsupervised or Semi-supervised-based** | | | | | | | | |
| B-factor | ✓ | 0.2078 | 0.1850 | 0.2009 | 0.2445 | 2.6557 | 2.0186 | 0.5876 |
| ESM-IF | ✓ | 0.2016 | 0.1491 | 0.3260 | 0.3353 | 2.6446 | 1.9555 | 0.6373 |
| MIF-$\Delta$logit | ✓ | 0.1053 | 0.0783 | 0.3358 | 0.2886 | 2.5361 | 1.8967 | 0.6066 |
| MIF-Net. | ✓ | 0.3968 | 0.3789 | 0.6139 | 0.5370 | 2.1399 | 1.6422 | 0.7735 |
| RDE-Linear | ✓ | 0.1763 | 0.2056 | 0.4583 | 0.4247 | 2.4460 | 1.8128 | 0.6573 |
| RDE-Net. | ✓ | 0.4233 | 0.3926 | 0.6288 | 0.5900 | 2.0980 | 1.5747 | 0.7749 |
| Refine-PPI | ✗ | 0.4474 | 0.4134 | 0.6307 | 0.5839 | 2.0939 | 1.589 | 0.7831 |
| Refine-PPI | ✗ | **0.4558** | **0.4289** | **0.6458** | **0.6091** | **2.0601** | **1.554** | **0.8064** |

## B.2 INITIALIZATION OF VARIANCE

we investigate three kinds of initialization mechanism for $\Sigma$. First and naively, we turn all $\Sigma_i$ into an identity matrix. Second, we depend on physical principles and utilize molecular dynamic (MD) simulations to attain the short motion trajectories (10 nanoseconds) of these complexes in the 3D space. Then we calculate the root-mean square fluctuation (RMSF) of each amino acid and take this value to initialize $\Sigma$. Third, we adopt a learnable strategy to initialize $\Sigma$. To be explicit, an embedding layer is created to each category of 20 residue types to a 3-dimension continuous vector. This routine learns the variance of different components completely from the data.

The performance of different initialization approaches are listed in Table 6, and it can be found that the constant initialization is the worst. In addition, the MD-based methodology slightly outperforms the embedding-based one. However, since MD simulations are time-consuming and costly, it is prohibited to implement MD during the inference stage each time. As a consequence, we use the third sort in our paper.

Table 5: Evaluation of $\Delta\Delta G$ prediction on the single-mutation subset of the SKEMPI.v2 dataset.

| Method | Pretrain | Per-Structure | | Overall | | | | |
| --- | --- | --- | --- | --- | --- | --- | --- | --- |
| | | Pearson | Spearman | Pearson | Spearman | RMSE | MAE | AUROC |
| **Energy Function-based** | | | | | | | | |
| Rosetta | – | 0.3284 | 0.2988 | 0.3113 | 0.3468 | 1.6173 | 1.1311 | 0.6562 |
| FoldX | – | 0.3908 | 0.3640 | 0.3560 | 0.3511 | 1.5576 | 1.0713 | 0.6478 |
| **Supervised-based** | | | | | | | | |
| DDGPred | ✗ | 0.3711 | 0.3427 | 0.6515 | 0.4390 | 1.3285 | 0.9618 | 0.6858 |
| End-to-End | ✗ | 0.3818 | 0.3426 | 0.6605 | 0.4594 | 1.3148 | 0.9569 | 0.7019 |
| **Sequence-based** | | | | | | | | |
| ESM-1v | ✓ | 0.0422 | 0.0273 | 0.1914 | 0.1572 | 1.7226 | 1.1917 | 0.5492 |
| PSSM | ✓ | 0.1215 | 0.1229 | 0.1224 | 0.0997 | 1.7420 | 1.2055 | 0.5659 |
| MSA Transf. | ✓ | 0.1415 | 0.1293 | 0.1755 | 0.1749 | 1.7294 | 1.1942 | 0.5917 |
| Tranception | ✓ | 0.1912 | 0.1816 | 0.1871 | 0.1987 | 1.7455 | 1.1708 | 0.6089 |
| **Unsupervised or Semi-supervised-based** | | | | | | | | |
| B-factor | ✓ | 0.1884 | 0.1661 | 0.1748 | 0.2054 | 1.7242 | 1.1889 | 0.6100 |
| ESM-IF | ✓ | 0.2308 | 0.2090 | 0.2957 | 0.2866 | 1.6728 | 1.1372 | 0.6051 |
| MIF-$\Delta$logit | ✓ | 0.1616 | 0.1231 | 0.2548 | 0.1927 | 1.6928 | 1.1671 | 0.5630 |
| MIF-Net. | ✓ | 0.3952 | 0.3479 | 0.6667 | 0.4802 | 1.3052 | 0.9411 | 0.7175 |
| RDE-Linear | ✓ | 0.3192 | 0.2837 | 0.3796 | 0.3394 | 1.5997 | 1.0805 | 0.6027 |
| RDE-Net. | ✓ | 0.4687 | 0.4333 | 0.6421 | 0.5271 | 1.3333 | 0.9392 | 0.7367 |
| Refine-PPI | ✗ | 0.4474 | 0.4134 | **0.6667** | **0.5338** | **1.2963** | **0.9179** | 0.7431 |
| Refine-PPI | ✓ | **0.4701** | **0.4459** | 0.6658 | 0.5153 | 1.2978 | 0.9287 | **0.7481** |

Table 6: Performance of different initialization methodologies for the coordinate variance $\Sigma$ (without pretraining).

| Method | Per-Structure | |
| --- | --- | --- |
| | Pearson | Spearman |
| Identity Matrix | 0.4422 | 0.4043 |
| MD Simulations | **0.4522** | **0.4287** |
| Learnable Variance | 0.4475 | 0.4102 |

## B.3 POSITION VARIANCE UPDATE IN PDC-EGNN

Notably, the way to update the variance of the positions of different atoms is not unique. Here, we offer another kind of approach to renew the variance in the layer of PDC-EGNN.

$$\Sigma_i^{(l+1)} = \left(1 + \frac{1}{|\mathcal{N}(i)|}\sum_{j\in\mathcal{N}(i)}\phi_\mu(\mathbf{m}_{j\to i})\right)^2 \Sigma_i^{(l)} + \frac{1}{|\mathcal{N}(i)|}\sum_{j\in\mathcal{N}(i)}\phi_\mu(\mathbf{m}_{j\to i})\Sigma_j^{(l)}, \quad (8)$$

where we leverage the same $\phi_\mu$ instead of a new $\phi_\sigma$. Besides, we distribute and square the $\mathbf{x}_i$ terms because $\mathbf{x}_i - \mathbf{x}_j$ is not independent of $\mathbf{x}_i$. Noticeably, this Equation 8 does not damage the equivariance property of our model. Experiments show that this form of position variance computation performs slightly better in the mutant effect prediction task (see Table 7), with a per-structure Spearman of 0.4490.

Table 7: Performance of different position variance update methods (without pretraining).

| Method | Per-Structure | |
| --- | --- | --- |
| | Pearson | Spearman |
| Equ. 7 | 0.4475 | 0.4102 |
| Equ. 8 | **0.4490** | **0.4153** |

### B.4 QUANTITATIVE ANALYSIS OF LEARNED UNCERTAINTY

In addition to the visualization of several complexes in Figrue 5, we quantitatively investigate the correlation between the learned variance and the positional uncertainty of the ground truth. To be specific, we run short MD simulations of each seed complex and compute the root mean square fluctuation (RMSF) of each amino acid. Additionally, we determine the magnitude of $||\Sigma_i||^2$, which is the learned variance of our uncertainty module. A detailed comparison of these values, classified by residues at and not at the interface, is presented in Table 8. Notably, the ground truth RMSF at the interface is significantly smaller than that observed elsewhere. At the same time, the learned $\Sigma_i$ exhibits a parallel pattern, where $||\Sigma_i||^2$ at the interface is much smaller. This quantitative analysis serves to substantiate the claim that the learned variance indeed corresponds to uncertainty.

Table 8: Performance of different position variance update methods (without pretraining).

|  | Interface | Non-Interface |
|---|---|---|
| RMSF | 0.4945 | **0.9735** |
| $||\Sigma_i||^2$ | 0.6072 | **0.8940** |

## C VISUALIZATION OF HALLUCINATED STRUCTURES

Here we provide some instances of mutant structures hallucinated by our Refine-PPI in Figure 9. Since the ground truth mutant structures are inaccessible, we leave it for future work to examine its accuracy.

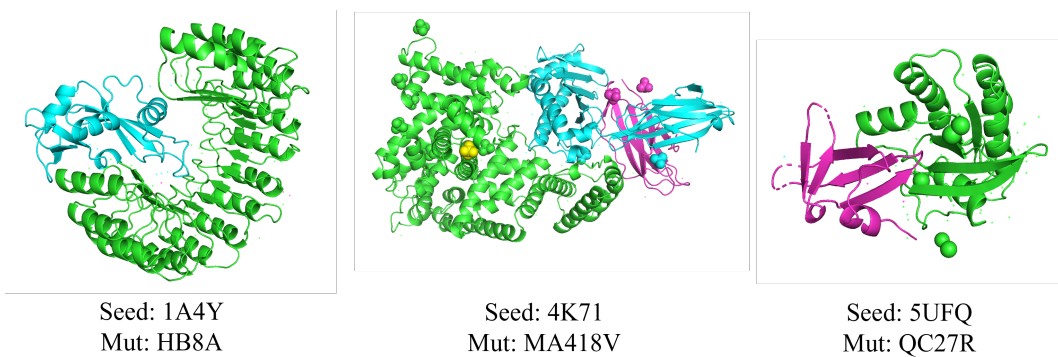

Seed: 1A4Y          Seed: 4K71          Seed: 5UFQ
Mut: HB8A          Mut: MA418V          Mut: QC27R

Figure 9: Examples of hallucinated structures of mutation-type.

## D PROOF OF EQUIVARIANCE

Equivariance is an important characteristic, and here, we demonstrate that PDC-Net strictly follow this rule of principle. More formally, for any translation vector $g \in \mathbb{R}^3$ and for any orthogonal matrix $Q \in \mathbb{R}^{3 \times 3}$, the model should satisfy:

$$\mathbf{h}^{(l+1)}, \left\{ Q\boldsymbol{\mu}_i^{(l+1)} + g, Q^\top \boldsymbol{\Sigma}_i^{(l+1)} Q \right\}_{i=1}^n = \text{PDC-L} \left[ \mathbf{h}^{(l)}, \left\{ Q\boldsymbol{\mu}_i^{(l)} + g, Q^\top \boldsymbol{\Sigma}_i^{(l)} Q \right\}_{i=1}^n, \mathcal{E} \right]. \quad (9)$$

We will analyze how translation and rotation of input coordinates propagate through our model. We start by assuming that $\mathbf{h}^0$ is invariant to the $\mathrm{E}(n)$ transformations on the co-ordinate distributions $\boldsymbol{\nu}$. In other words, information on the absolute position or orientation of $\boldsymbol{\nu}^0$ is not encoded in $\mathbf{h}^0$. Then, the distance between two particles is invariant to translations, rotations and reflections. This is because, for the mean of distance $\mu_{d_{ij}}$, we have $\text{tr}\left( Q^\top \boldsymbol{\Sigma}_i Q + Q^\top \boldsymbol{\Sigma}_j Q \right) = \text{tr}\left( \boldsymbol{\Sigma}_i + \boldsymbol{\Sigma}_j \right)$ due to the characteristic of the isotropic matrix and

$||Q\boldsymbol{\mu}_i^{(l)} + g - (Q\boldsymbol{\mu}_j^{(l)} + g)||^2 = ||Q\boldsymbol{\mu}_i^{(l)} - Q\boldsymbol{\mu}_j^{(l)}||^2 = (\boldsymbol{\mu}_i^{(l)} - \boldsymbol{\mu}_j^{(l)})^\top Q^\top Q(\boldsymbol{\mu}_i^{(l)} - \boldsymbol{\mu}_j^{(l)}) = (\boldsymbol{\mu}_i^{(l)} - \boldsymbol{\mu}_j^{(l)})^\top \mathbf{I}(\boldsymbol{\mu}_i^{(l)} - \boldsymbol{\mu}_j^{(l)}) = ||\boldsymbol{\mu}_i^{(l)} - \boldsymbol{\mu}_j^{(l)}||^2$. Meanwhile, for the variance of distance $\sigma_{d_{ij}}$, we have $[Q\boldsymbol{\mu}_i + g - (Q\boldsymbol{\mu}_j + g)]^\top (Q^\top \boldsymbol{\Sigma}_i Q + Q^\top \boldsymbol{\Sigma}_j Q) [Q\boldsymbol{\mu}_i + g - (Q\boldsymbol{\mu}_j + g)] = (\boldsymbol{\mu}_i - \boldsymbol{\mu}_j)^\top Q^\top (\boldsymbol{\Sigma}_i + \boldsymbol{\Sigma}_j) Q(\boldsymbol{\mu}_i - \boldsymbol{\mu}_j) = (\boldsymbol{\mu}_i - \boldsymbol{\mu}_j)^\top (\boldsymbol{\Sigma}_i + \boldsymbol{\Sigma}_j) (\boldsymbol{\mu}_i - \boldsymbol{\mu}_j)$. Consequently, the output $\mathbf{m}_{j \rightarrow i}$ will also be invariant as the edge operation $\phi_e(.)$ becomes invariant.

Aterwards, the equations of our model that update the mean and variance of coordinates $\mathbf{x}$ are $\mathrm{E}(n)$ equivariant as well. In the following, we prove their equivariance by showing that a $\mathrm{E}(n)$ transformation of the input leads to the same transformation of the output. Notice that $\mathbf{m}_{j \rightarrow i}$ is already invariant as proven above. Notably, the translation $g$ has no impact over the variance of coordinates $\boldsymbol{\Sigma}_i^{(l)}$. Thus, we want to show:

$$Q\boldsymbol{\mu}_i^{(l+1)} + g = Q\boldsymbol{\mu}_i^{(l)} + g + \frac{1}{|\mathcal{N}(i)|} \sum_{j \in \mathcal{N}(i)} \left( Q\boldsymbol{\mu}_i^{(l)} + g - \left[ Q\boldsymbol{\mu}_j^{(l)} + g \right] \right) \phi_\mu(\mathbf{m}_{j \rightarrow i}),$$

$$Q^\top \boldsymbol{\Sigma}_i^{(l+1)} Q = Q^\top \boldsymbol{\Sigma}_i^{(l)} Q + \frac{1}{|\mathcal{N}(i)|} \sum_{j \in \mathcal{N}(i)} \left( Q^\top \boldsymbol{\Sigma}_i^{(l)} Q + Q^\top \boldsymbol{\Sigma}_j^{(l)} Q \right) \phi_\sigma(\mathbf{m}_{j \rightarrow i}).$$

(10)

Its derivation is as follows.

$$Q\boldsymbol{\mu}_i^{(l)} + g + \frac{1}{|\mathcal{N}(i)|} \sum_{j \in \mathcal{N}(i)} \left( Q\boldsymbol{\mu}_i^{(l)} + g - \left[ Q\boldsymbol{\mu}_j^{(l)} + g \right] \right) \phi_\mu(\mathbf{m}_{j \rightarrow i})$$

$$= Q\boldsymbol{\mu}_i^{(l)} + g + Q\frac{1}{|\mathcal{N}(i)|} \sum_{j \in \mathcal{N}(i)} \left( \boldsymbol{\mu}_i^{(l)} - \boldsymbol{\mu}_j^{(l)} \right) \phi_\mu(\mathbf{m}_{j \rightarrow i})$$

$$= Q \left( \boldsymbol{\mu}_i^{(l)} + \frac{1}{|\mathcal{N}(i)|} \sum_{j \in \mathcal{N}(i)} \left( \boldsymbol{\mu}_i^{(l)} - \boldsymbol{\mu}_j^{(l)} \right) \phi_\mu(\mathbf{m}_{j \rightarrow i}) \right) + g$$

$$= Q\boldsymbol{\mu}_i^{(l+1)} + g.$$

(11)

$$Q^\top \boldsymbol{\Sigma}_i^{(l)} Q + \frac{1}{|\mathcal{N}(i)|} \sum_{j \in \mathcal{N}(i)} \left( Q^\top \boldsymbol{\Sigma}_i^{(l)} Q + Q^\top \boldsymbol{\Sigma}_j^{(l)} Q \right) \phi_\sigma(\mathbf{m}_{j \rightarrow i})$$

$$= \boldsymbol{\Sigma}_i^{(l)} + \frac{1}{|\mathcal{N}(i)|} \sum_{j \in \mathcal{N}(i)} \left( \boldsymbol{\Sigma}_i^{(l)} + \boldsymbol{\Sigma}_j^{(l)} \right) \phi_\sigma(\mathbf{m}_{j \rightarrow i})$$

(12)

$$= \boldsymbol{\Sigma}_i^{(l+1)} = Q^\top \boldsymbol{\Sigma}_i^{(l+1)} Q.$$

Therefore, we have proven that rotating and translating the mean and variance of $\mathbf{x}^{(l)}$ results in the same rotation and translation on the mean and variance of $\mathbf{x}^{(l+1)}$.

Furthermore since the update of $\mathbf{h}^{(l)}$ only depend on $\mathbf{m}_{j \rightarrow i}$ and $\mathbf{h}^{(l)}$ which as saw at the beginning of this proof, are $\mathrm{E}(n)$ invariant, therefore, $\mathbf{h}^{(l+1)}$ will be invariant too. Thus, we conclude that a transformation $Q\boldsymbol{\mu}_i^{(l)} + g$ in $\boldsymbol{\mu}_i^{(l)}$ will result in the same transformation on $\boldsymbol{\mu}_i^{(l+1)}$ while $\mathbf{h}^{(l+1)}$ will remain invariant to it such that $\mathbf{h}^{(l+1)}, \left\{ Q\boldsymbol{\mu}_i^{(l+1)} + g, Q^\top \boldsymbol{\Sigma}_i^{(l+1)} Q \right\}_{i=1}^n =$ PDC-L $\left[ \mathbf{h}^{(l)}, \left\{ Q\boldsymbol{\mu}_i^{(l)} + g, Q^\top \boldsymbol{\Sigma}_i^{(l)} Q \right\}_{i=1}^n, \mathcal{E} \right]$ is satisfied.

## E  LIMITATIONS AND FUTURE WORK

In spite of the success of Refine-PPI in estimating the mutation effect, there are still rooms left for improvements. First, Refine-PPI keeps most of the complex stable and merely restores a region around the mutant site. It is possible that the enire complex can be significantly different upon the mutation. Therefore, a promsing future direction would be enlarge the mask region. Besides, preceding studies demonstrate the benefit of structural pretraining to dramatically expand the representation space of DL models. We expect to implement MMM with more experimental structures other than PDB (*e.g.*, Alphafold-Database) and transfer the knowledge to predict free energy change.

