# OpenReview forum: "Thermodynamics-inspired Structure Hallucination for Protein-protein Interaction Modeling"
_ICLR.cc/2024/Conference — Submitted to ICLR 2024_

### Official Review · Reviewer_Criu · 2023-10-24

**Soundness:** 2 fair
**Presentation:** 3 good
**Contribution:** 2 fair
**Rating:** 5
**Confidence:** 4

**Summary:**

The paper deals with the problem of mutation stability prediction—i.e., given a mutation to a bound complex with known structure, predicting the change in binding free energy. The authors make two contributions (1) they propose _masked mutation modeling_ (MMM) a auxiliary training task where the model must generate a structure for the mutant complex, and this generated (“hallucinated”) structure is used as additional input to the free energy predictor; (2) they introduce a _probability density cloud_ (PDC) modification to EGNN which is meant to capture the uncertainty in atomic positions. The empirical results slightly improve over RDE, the previous state-of-the-art.

**Strengths:**

* The introduction of the MMM task is quite sensible and well-suited for data-poor tasks such as mutation stability prediction.
* The “probability density cloud” modification to EGNN is quite interesting and represents a commendable attempt to introduce physical inductive biases into point cloud networks.
* The clarity of the exposition is above average for ICLR submissions. Each contribution is well-motivated and contextualized.

**Weaknesses:**

* The PDC module, while a very interesting idea, is on shaky ground mathematically.
    * The authors assume all atom positions to be independent, which is a very questionable assumption as the thermal fluctuations and epistemic uncertainty of neighboring atoms certainly should be dependent.
    * It is not possible to write down the self-covariance of the difference of random variables without knowing the cross-covariances. However, even assuming the positions to be independently distributed, then if their mean is updated as in Eq 6, then Eq 7 should read $${\sigma_{x_i}^{(l+1)}}^2 = \left[1 + \frac{1}{|N(i)|}\sum_{j \in N(i)} \phi_\mu(m_{j \rightarrow i})\right]^2{\sigma_{x_i}^{(l)}}^2 + \frac{1}{|N(i)|}\sum_{j \in N(i)}{\sigma_{x_j}^{(l)}}^2\phi_\mu(m_{j \rightarrow i})$$ i.e., with $\phi_\mu$ instead of a different $\phi_\sigma$, and distributing and squaring the $x_i$ terms because $x_i - x_j$ is not independent of $x_i$.
    * With that said, I don’t think it’s a serious issue in itself if the network is not updating the co-variances properly. But my general concern is that the authors have not given sufficient treatment to these subtleties, and hence, the PDC module is not actually constrained to model the atomic uncertainties as the authors claim; rather, to the network ${\sigma_{x_i}^{(l+1)}}^2$ just looks like some other latent variable which it can use to help model the positional updates. It would be better to call the module “loosely inspired” by the modeling of uncertainty. If the authors nevertheless claim that this is a generally helpful modification to EGNN, this is a claim that requires significantly more thorough evaluation (on many different tasks ideally) than is given here.

* There are several critical missing details in the methodology and experiments (see Questions below).

* Some over- or mis-claiming throughout the paper
    * The authors claim (Eq 1) to recover the structural distribution of the masked residues, but the training objective is risk minimization (Eq 3), not any kind of distributional modeling objective.
    * The authors state that MMM “encourages graph manifold learning with the denoising objective”, but there is no further discussion or elaboration on how “graph manifold learning” is accomplished.
    * Repeatedly misleading use of the term “thermodynamics” when the authors mean “uncertainty.” The former term should be reserved when explicitly referring to physically meaningful quantities like energy, entropy, and free energy.
    * “The pictures show that particles in the interface have smaller variation compared to those in the edges of proteins.” This claim is not backed quantitatively, and, as discussed, there is no reason to believe that the learned $\sigma^2$ actually corresponds to uncertainty.
    * “It can be found that generally, a small error of wide-type structure reconstruction leads to a more accurate $\Delta\Delta G$ prediction.” I see no such correlation in Figure 4B.



Justification for score: there is the potential for interesting technical contribution in the PDC, but the current presentation is not thorough enough for a conference paper. The MMM objective by itself is less novel, as auxiliary training or pretraining for mutational stability prediction has been done before, and the results are only a bit better than those prior approaches.

**Questions:**

* Methodology details and design are unclear
    * Where is the PDC module actually used? How are the $\sigma^2$s initialized?
    * “Moreover, it is readily apparent that PDC-Net maintains the equivariance property.” The authors should provide a proof here.
    * Because there are no gradients from the $\Delta\Delta G$ task to the structure refinement module $f_\theta$, the MMM task is really a pretraining task and not an auxiliary training task. Is there any reason to not train MMM across the entire PDB?
    * Is there any reason to use the same encoding module $h_\rho$ for the $\Delta\Delta G$ predictor and for the structure refinement module $f_\theta$. Why can’t $f_\theta$ carry its own encoding module?
* Experimental details
    * It is not clear how the baselines are run in order to obtain \Delta\Delta G predictions, especially. ESM-1v, B-Factor, etc. While some reasonable guesses exist, the authors should spell it out and not leave it up to guessing.
    * How many different complexes are in SKEMPI and is the average per-complex improvement statically significant?
    * In the ablation studies, why does Model 1 use the RDE-Net backbone instead of the Refine-PPI modules $h_\rho$?
* Minor issues
    * Broken link to figure 3.2 where a visualization of the PDCs is promised.
    * The term “probability density cloud” suggests a more expressive parameterization than Gaussian uncertainty. I suggest the authors rename the module.
    * In Table 1, what is the meaning of “pretraining”? How is it possible that Refine-PPI and ESM are classified as no-pretraining, yet B-factors are classified as pretraining?
    * Inconsistent use of $\Sigma$ vs $\sigma$.
    * Typos: “Wide type”, "paradiagm", "disucssion", "intergrate", "envision" instead of "visualize"
    * The clarity could be improved with a figure illustrating the coordinate initialization.

---

> ### Author Response · Authors · 2023-11-13
> **Response to Reviewer Criu**
>
> Reviewer Criu, we appreciate your thoughtful and detailed review of our paper on mutation stability prediction. Your constructive feedback has been instrumental in identifying areas for improvement. We acknowledge your positive feedback on the introduction of the MMM task and the PDC modification to EGNN and also appreciate your recognition of the clarity in the exposition of our contributions. Below, please allow us to address each of your comments and concerns in turn.
>
> **Questions**
>
> (1) Methodology Details
>
> -- To elucidate, the backbone module $h_\rho$ can take the form of any conventional geometric neural network (e.g., GVP-GNN, EGNN, SE(3)-Transformer, Graph-Transformer). Here, we adopt a one-layer Graph Transformer [A, B] to extract general representations of proteins. The refinement model $f_\theta$ needs to output both updated features and coordinates, and, therefore, **we use the PDC module as $f_\theta$** in our experiments. Lastly, the head predictor $g_\tau$ is a simple linear layer that accepts the concatenation of both wide-type and mutation-type representations and forecasts the free energy change. The total model size of Refine-PPI is approximately 6M.
>
> [A] Shan, Sisi, et al. "Deep learning guided optimization of human antibody against SARS-CoV-2 variants with broad neutralization." Proceedings of the National Academy of Sciences 119.11 (2022): e2122954119.
>
> [B] Luo, Shitong, et al. "Antigen-specific antibody design and optimization with diffusion-based generative models for protein structures." Advances in Neural Information Processing Systems 35 (2022): 9754-9767.
>
> Specifically, we investigate three sorts of initialization mechanisms for $\sigma$. First and naively, we turn all $\sigma$ to be equivalent to one (a unit length). Second, we depend on physical principles and utilize molecular dynamic (MD) simulations to attain the short motion trajectories (10 nano-seconds) of these complexes in the 3D space. Then we calculate the root-mean-square fluctuation (RMSF) of each amino acid and take this value as the initial input of $\sigma$. Third, we adopt a learnable strategy to initialize $\sigma$. To be explicit, an embedding layer is created for each category of 20 residue types to a 3-dimensional continuous vector. This routine learns the variance of different components completely from the data.
>
> The performance of different initialization approaches is listed below, and it can be found that constant initialization is the worst. Besides, the MD-based methodology outperforms slightly better results than the embedding-based one. However, since MD simulations are time-consuming and costly, it is prohibited to implement MD during the inference stage each time. As a consequence, we use the third sort of initialization method in our paper. In the revised manuscript, we provide a more detailed explanation of the initialization process in the Appendix.
> |       Method       | Per-Structure | Per-Structure |
> |:------------------:|:-------------:|:-------------:|
> |                    |    Pearson    |    Spearman   |
> |     Unit Length    |     0.4422    |     0.4043    |
> |   MD Simulations   |    **0.4522**   |   **0.4287**  |
> | Learnable Variance |     0.4475    |     0.4102    |
>
> -- Thanks for your advice. We have provided proof in the Appendix. It would be helpful for readers who are interested in the equivariance property of our model.
>
> -- Thanks for your question about the gradient backpropagation. Notably, since the MMM task and the $\Delta\Delta G$ task share the same encoder $h_\rho$, the MMM task here is an auxiliary training task instead of a pretraining task. We have tried to allow gradient backpropagation from the $\Delta\Delta G$ task to the structure refinement module $f_\theta$ but found the training process unstable with slightly worse performance. This phenomenon can be attributed to the small number of available experimental data in the downstream Skempi v2 dataset. It can be expected that if given more ground truth complex structures, the training procedure would be stabilized and the allowance of gradient backpropagation can lead to enhanced results.

---

> ### Author Response · Authors · 2023-11-13
> **Response to Reviewer Criu (Part II)**
>
> Yes! you are absolutely correct that MMM can be trained across the entire PDB rather than the limited 345 wide-type structures. Our preliminary aim is to propose a new framework for **solving the absence of mutant structures with no additional database such as PDB and Alphafold-DB**. Moreover, we have already observed that Refine-PPI outperforms existing algorithms without any pretraining technique. Therefore, we only report the non-pretrained performance in our paper. However, it is undoubtedly expected to have a strong promotion on the capability of our Refine-PPI once it is pretrained on PDB, and please see the updated results below. It can be found that **the per-structure Spearman correlation will significantly increase from 0.41 to 0.44**. Notably, We emphasize the importance of per-structure metrics over overall metrics for practical applications.
>
> |   Method   | Pretrain | Per-Structure | Per-Structure |   Overall  |   Overall  |   Overall  |   Overall  |   Overall  |
> |:----------:|:--------:|:-------------:|:-------------:|:----------:|:----------:|:----------:|:----------:|:----------:|
> |            |          |    Pearson    |    Spearman   |   Pearson  |  Spearman  |    RMSE    |     MAE    |    AUROC   |
> |   RDE-Net  |    Yes   |     0.4448    |     0.4010    |   0.6447   |   0.5584   |   0.5799   |   1.1123   |   0.7454   |
> | Refine-PPI |    No    |     0.4475    |     0.4102    |   0.6584   |   0.5394   | **1.5556** | **1.0946** |   0.7517   |
> | Refine-PPI |    Yes   |   **0.4561**  |   **0.4374**  | **0.6592** | **0.5608** |   1.5643   |   1.1093   | **0.7542** |
>
> -- Thanks for your question regarding the design of our Refine-PPI. To be specific, we adopt the same encoder $h_\rho$ for both the $\Delta\Delta G$ predictor and structure refinement module $f_\theta$, so that the knowledge learned by $f_\theta$ in structure prediction can be jointly used by the $\Delta\Delta G$ predictor. We hold the view that the capacity to be aware of geometries in mutant regions is of great significance for deep learning models to predict mutant effects, especially when mutant structures are absent. If we make $f_\theta$ to carry its own encoding module, then the MMM and $\Delta\Delta G$ prediction are two completely independent tasks and will not enjoy the benefit of multi-task learning.
>
> (2) Experimental Details
>
> -- Thanks for your suggestions about the baseline implementations. It is absolutely very necessary to spell them out rather than leave it up to guessing. We have formally stated the details in the Appendix about how these baselines are run to obtain $\Delta\Delta G$.
>
> -- There are 345 different complexes in Skempi V2, and the average per-complex improvement is statistically significant. Please see Section B in the Appendix, where we envision some case studies in SKMEMPI and select 4 examples for each different number of mutations.
>
> -- We apologize for the confusion in Model 1. In our experiments, the backbone module $h_\rho$ is exactly the same as the RDE-Net backbone. Thus, we adopt the RDE-Net backbone for Model 1 in the ablation studies for fair comparison.
>
> (3) Minor Issues
>
> -- We have rectified the broken link to Figure 3.2 by using '\hyperref'. Thanks for pointing out this issue!
>
> -- We agree that the “probability density cloud” suggests a more expressive parameterization. Do you think it would be better and properer if we adopt 'Gaussian Uncertainty Network' or 'Uncertainty-aware Neural Network' to name our module?
>
> -- We acknowledge your concern about the meaning of "pretraining". We classify baselines and our approach as pretrained or non-pretrained by the criteria of whether they have been trained on or use any additional database such as PDB and Alphafold-DB. Traditionally, "pretraining" is usually applied to deep learning models. Since Rosetta and FoldX are energy-based methods, we leave them alone. After more careful consideration, we separate ESM, PSSM, MAS-Transformer, Tranception, B-factor, MIF, and RDE as pretrained methods, while regarding DDGPred and End-to-End as non-pretrained. Notably, B-factor is first pretrained to predict per-atom b-factors for proteins in PDB and then is transferred to this mutant effect prediction task. We have updated the latest classification of those baselines in the revised version. Great thanks for helping and encouraging us to clarify this issue!
>
> -- We appreciate your clarification on the usage of $\Sigma$ and $\sigma$. The main reason is that $\boldsymbol{\Sigma}_i\in \mathbb{R}^{3\times 3}$ is a diagonal covariance matrix indicating that different axes are independent of each other. Consequently, we utilize $\boldsymbol{\sigma}$ to denote the diagonal elements of the variance matrix $\mathbf{\Sigma}$ for simplicity. we will incorporate this explanation into our revised manuscript.

---

> ### Author Response · Authors · 2023-11-13
> **Response to Reviewer Criu (Part III)**
>
> -- Thanks for your advice for improving the clarity. We have added a figure in the Appendix to illustrate the coordinate initialization. We believe with this new figure, the audience can understand the MMM task more easily.
>
> **The design of PDC-Net**
>
> -- We appreciate your diligence in scrutinizing the mathematical foundation of our PDC module and also acknowledge that the thermal fluctuations and epistemic uncertainty of neighboring atoms certainly should be dependent. However, if we assume that the positions of particles are highly correlated to their neighbors, the mathematical analysis of their uncertainty becomes intricate. Just as you mentioned, we first need to hypothesize the cross-covariance of each pair of entities in the molecular system so that the self-covariance of each atom can be calculated. A major challenge lies in computing geometric variables like distances and angles. For instance, once $\mathbf{x}_i$ and $\mathbf{x}_j$
>
> follow dependent Gaussian distributions, the squared norm of their difference $d_{ij}$  will deviate from a generalized chi-square distribution $\chi^2(.)$. Due to our mathematical limitations, explicitly defining a set of natural parameters to model the exact distribution of $d_{ij}$ is not feasible. Additionally, portraying higher-order geometries, such as the angle between vectors $\mathbf{x}_i - \mathbf{x}_j$ and $\mathbf{x}_i - \mathbf{x}_k$, becomes more challenging. We really understand and align with your perspective that close atoms have a significant impact on each other in the 3D space, and plan to address this in future work by introducing a more nuanced architecture. It is undeniable that separating individual particles into isolated elements provides great convenience for us to model complex molecular systems. More importantly, the interdependency between close entities will be captured via the following message-passing-based mechanism when training geometric networks.
>
> -- Thanks for your advice on the update of coordinate variance! It is rational to use the same operation $\phi_\mu$ for renewing both mean and variance rather than employing separate operations (i.e., $\phi_\mu$ and $\phi_\sigma$). In addition, we concur with your suggestion to square the $\mathbf{x}_i$ terms since the vector $\mathbf{x}_i - \mathbf{x}_j$ is dependent on $\mathbf{x}_i$. Experiments show that this form of position variance computation performs slightly better in the mutant effect prediction task (without pretraining on PDB). We have included this new Equation in the Appendix of our revised manuscript and would like to express our gratitude for your insight on PDC-EGNN. We believe the introduction of positional uncertainty is a promising direction, worthy of further exploration and the proposal of advanced architectures.
> |  Method  | Per-Structure | Per-Structure |
> |:--------:|:-------------:|:-------------:|
> |          |    Pearson    |    Spearman   |
> |  Equ. 7  |     0.4475    |     0.4102    |
> | New Equ. |     **0.4490**    |    **0.4153**    |
>
> -- We agree with your assertion that modeling atomic uncertainty is crucial, prompting us to consider a more fitting name for our PDC module. As suggested, we are contemplating renaming it as the **"Uncertainty-aware Neural Network"** or **"Gaussian Uncertainty Network."** Do you think this modification is better? We are open to your thoughts on whether this modification is preferable or if an "Uncertainty-inspired Neural Network," as mentioned in your comment, would be a more suitable choice.
>
> **Over- or Mis-claiming throughout the Paper**
>
> -- Thanks for your question regarding the structural distribution recovery. You are right that our aim is to recover the structural distribution of the masked residues, namely, $p(\{\mathbf{x}^{\textrm{WT}}\}^{m+r}_{i=m-l} \big| \tilde{\mathcal{G}}^{\textrm{WT}}, a_m, \theta, \rho )$.
>
> In order to supervise the MMM task, the exact training object would be measuring and minimizing the gap between the predicted structural distribution $Q$ and the ground truth structural distribution $P$. A widely used measure is the KL-divergence, writtern as $D_{KL}(P||Q) = \mathbb{E}_{x\sim P}\big[log \frac{P(X)}{Q(X)}\big]$. Then if our goal is to learn a model $f:\mathcal{X} \rightarrow \mathcal{Y}$, our objective becomes
>
> ${\textrm{argmin}}_ \theta  \mathbb{E}_ {x,y\sim\mathcal{D}} [-logQ_ \theta (y|x)].$
>
> For classification problems, the cross-entropy loss is exactly what KL divergence minimizes. On the contrary, for regression task, minimizing the negative log likelihood (NLL) of this normal distribution is clearly equivalent to the mean-squared-error loss, namely, Equ.3 in our paper. For more information, please refer to this wonderful blog: https://dibyaghosh.com/blog/probability/kldivergence.html.

---

> ### Author Response · Authors · 2023-11-13
> **Response to Reviewer Criu (Part IV)**
>
> -- We appreciate your clarification on the use of terminology. Please allow us to address the point you raised about the claims related to the graph manifold learning. To be specific,  By using an implicit mapping from corrupted data to clean data, the MMM objective encourages the model to learn the manifold on which the clean data lies — we speculate that the deep learning model learns to go from low probability graphs to high probability graphs. In the denoising-based initialization case, Refine-PPI learns the manifold of the input data. When node targets are provided, the model learns the manifold of the target data (e.g. the manifold of atoms at equilibrium). We speculate that such a manifold may include commonly repeated substructures that are useful for downstream prediction tasks. A similar motivation can be found in prior denoising-based molecular pretraining methods [A, B, C].
>
> [A] Godwin, Jonathan, et al. "Simple gnn regularisation for 3d molecular property prediction & beyond." ICLR 2022.
>
> [B] Vincent, Pascal, et al. "Stacked denoising autoencoders: Learning useful representations in a deep network with a local denoising criterion." Journal of machine learning research 11.12 (2010).
>
> [C] Song, Yang, and Stefano Ermon. "Generative modeling by estimating gradients of the data distribution." Advances in neural information processing systems 32 (2019).
>
> --  We appreciate your guidance on the distinction between "thermodynamics" and "uncertainty." We will rectify the repeated misuse of the term "thermodynamics" in situations where "uncertainty" is more appropriate. This correction will ensure a more accurate and precise representation of our concepts.
>
> -- -- We acknowledge your concern about the need for quantitative support for certain claims, particularly in relation to the variation of particles at the protein interface. To address this, we conducted short trajectories of all structures in Skempi V2, calculating the Root Mean Square Fluctuation (RMSF) of each residue. Additionally, we determined the magnitude of $||\boldsymbol{\sigma}_{\mathbf{x}_i}||^2$, which is the learned variance of our uncertainty module. A detailed comparison of these values, categorized by residues at and not at the interface, is presented in the table below. Notably, the ground truth RMSF at the interface is significantly smaller than that observed elsewhere.
>
> Concurrently, the learned  $\boldsymbol{\sigma}$ exhibits a parallel pattern, where $||\boldsymbol{\sigma}_{\mathbf{x}_i}||^2$ at the interface is much smaller. This quantitative analysis serves to substantiate the claim that the learned variance indeed corresponds to uncertainty. We appreciate your suggestion to quantify the uncertainty, as it has enhanced the rigor of our analysis!
>
> |     | Interface | Non-Interface |
> |:-----------------------------:|:---------:|:-------------:|
> |     RMSF in MD Simulations    |   0.4945  |     0.9735    |
> | Magtitude of Learned Variance |   0.6072  |     0.8940    |
>
> -- We acknowledge the discrepancy you pointed out regarding the statement on a correlation between a small error in wide-type structure reconstruction and a more accurate prediction in Figure 4B. Upon further examination, we realize the oversight in the interpretation. We will detele the corresponding text for clarity and accuracy.
>
> -------------------------
>
> At last, we sincerely appreciate your commitment to maintaining the scientific rigor of our work, and your feedback will undoubtedly contribute to enhancing the quality and clarity of our manuscript. If you have any further comments or suggestions, please feel free to share them. Thank you for your time and consideration!

---

> ### Comment · Reviewer_Criu · 2023-11-15
>
> Thanks for the response. I appreciate the diligent resolution of many miscellaneous concerns. I also appreciate that the authors have tried out the suggested change to Eq 7. However:
>
> (1) I think the paper still lacks a clean and satisfactory conceptual presentation of the PDC module. I emphasize that I really like this direction of thinking, but it seems that the design and conceptual framework of such a module has not been carefully thought through and the current presentation raises more questions / confusion than it inspires answers.
>
> (2) I disagree with the claim that RMSE is a distributional modeling objective, because this is true only if the distributional model is Gaussian. It's actually not clear to me why it's necessary to make this claim, because it doesn't seem to affect the rest of the paper.
>
> (3) While there are cherry-picked examples to illustrate that the learned "variance" semantically corresponds to positional uncertainty, there is still, at the end of the day, no conceptual justification for believing that this is true systematically. Do the authors claim that any latent variable which is additive according to Eq 7 must be some kind of uncertainty? Note that while a loss term is applied directly on the learned positions, there is no loss term to enforce any kind of semantic meaning to the learned variances.
>
> For these reasons, I intend to keep the current score.

---

> > ### Author Response · Authors · 2023-11-16
> > **Reply to Official Comment of Reviewer Criu**
> >
> > Thank you for your thoughtful feedback on our paper. We are glad to hear that you are satisfied with some of our updates on the implementation details and additional experiments. Your insights are invaluable to us, and please see the reply to your concerns as below.
> >
> > (1) Regarding your first point about the conceptual presentation of the PDC module, we acknowledge the importance of a clear and satisfactory explanation. We will revisit the section and make revisions to ensure that the design and conceptual framework of the module are more meticulously articulated, aiming to reduce any ambiguity and address the questions you raised.
> >
> > (2) Upon further consideration, we agree with your point that RMSE should not be used as a distributional modeling objective. We reformulate our claim of MMM's training objective and state that "our aim is to recover the structure of this masked region" instead of "recover the structural distribution". Consequently, the entire process is written as follows.
> >
> > $f_\theta(\Tilde{\mathbf{Z}^{\textrm{MT}}}, \Tilde{\mathcal{G}}^{\textrm{WT}}, a _m) \rightarrow \{\mathbf{x}^{\textrm{WT}}\}^{m+r} _{i=m-l}.$
> >
> > As you said, this does not affect the rest of the paper, but we appreciate your rigorous attitude toward this statement in the paper. Any equations or mathematical claims should be examined to ensure their clarity and correctness. Thanks!
> >
> > (3) We really understand your concern about the justification for the correspondence between learned "variance" and positional uncertainty. We agree that our loss term primarily influences output positions without directly enforcing the network to capture uncertainty information.  However, it's essential to note that our PDC-module design (the final name is to be determined) is theoretically grounded in the concept of atomic uncertainty. To elucidate, posit that all atoms adhere to a Gaussian distribution and derive the geometric attributes such as distance and angles, which are also represented as distributions. Then our PDC-module is proposed to allow the propagation of these distributions and update their natural parameters during training. From this standpoint, we firmly believe **this latent variance $\Sigma$ corresponds to and at least encodes a form of uncertainty**.
> >
> > We acknowledge your desire for straightforward proof that exhibits the strong correspondence between our proposed "variance" and uncertainty. However, experimentally observing and documenting particle uncertainty within macromolecules, such as proteins, is exceedingly challenging, if not impossible. All existing data in PDB or Skempi v2 are deterministic and uncertainty-free conformations, making it difficult to quantitatively measure uncertainty and evaluate our architecture's effectiveness in capturing atomic uncertainty.
> >
> > As a solution, we resort to computational methods like molecular dynamics (MD) simulations to simulate the atomic motions. Notably, MD simulations approximate atomic motions by Newtonian physics and can capture the sequential behavior of molecules in full atomic details at a very fine temporal resolution, quantifying how much various regions of the molecule move at equilibrium and what types of fluctuations they undergo. We run short-time MD simulations for all complex structures and calculate the Root Mean Square Fluctuation (RMSF) alongside the entire trajectory. It is worth mentioning that **RMSF is a numerical measurement similar to RMSD, but instead of indicating positional differences between entire structures over time. It is a calculation of individual residue flexibility, or how much a particular residue moves (fluctuates) during a simulation [A, B]. Therefore, we use RMSF as a measurement of the atomic uncertainty**, and then navigate three approaches to demonstrate the capability of our PDC module to encode uncertainty using this MD-based data.
> >
> > First, we explore three sorts of variance initialization mechanisms. Specifically, i) all $\Sigma$ are identity matrices; ii) $\Sigma$ are learnable; iii) $\Sigma$ are initialized by RMSF from MD simulations. Results show that MD-based initialization achieves the best Spearman (0.4287), outweighing the learnable one (0.4102) and identity matrix (0.4043), emphasizing the efficacy of incorporating simulated uncertainty into the PDC module. This implies that simulated uncertainty is the optimal choice for this variance, and learned variance ideally should move towards this simulated uncertainty.

---

> ### Author Response · Authors · 2023-11-16
> **Reply to Official Comment of Reviewer Criu (Part II)**
>
> Second, since we have the simulated uncertainty, it is feasible for us to compute its similarity to learned variance. Specifically, we compare the magnitude of $||\boldsymbol{\sigma}_{\mathbf{x}_i}||^2$ with this simulated uncertainty, categorized by residues at and not at the interface. It can be found that both the simulated uncertainty and learned variance at the interface are significantly smaller than that observed elsewhere (presented in the Table below). This provides further evidence that our learned variance accords with a similar pattern of simulated uncertainty.
>
> |     | Interface | Non-Interface |
> |:-----------------------------:|:---------:|:-------------:|
> |     Simulated Uncertainty (RMSF)   |   0.4945  |     0.9735    |
> | Magtitude of $\boldsymbol{\sigma} _{\mathbf{x} _i} $ |   0.6072  |     0.8940    |
>
> Third, we implement additional experiments to directly predict the simulated uncertainty and examine its effectiveness. On the one hand, we adopt an EGNN with the PDC module and directly enforce the learnable variance to accord with the simulated uncertainty. The loss term is therefore set as $MSE(||\boldsymbol{\sigma}_{\mathbf{x}_i}||^2, RMSF)$.  On the other hand, we leverage a naive EGNN without the PDC module and require it to output RMSF based on the residue feature of the final layer. The loss is written as $MSE(MLP(\mathbf{h}^{(L)}), RMSF)$, where MLP is the abbreviation of the multi-layer perceptron. The experiments show that the PDC module significantly improves the capability of EGNN to forecast the simulated uncertainty. This phenomenon illustrates that our design of $\Sigma$ can be a good choice to represent and encode atomic uncertainty in the 3D space.
>
> |  Model   | MSE |
> |:-----------------------------:|:---------:|
> |     EGNN   |   0.2609  |
> | PDC-EGNN |   0.0381  |
>
> [A] RMSF Analysis. https://ctlee.github.io/BioChemCoRe-2018/rmsd-rmsf/#:~:text=RMSF%20stands%20for%20root%20mean,(fluctuates)%20during%20a%20simulation.
>
> [B] Wiki. https://en.wikipedia.org/wiki/Root-mean-square_deviation_of_atomic_positions
>
> To summarize, based on these three facts, we have the confidence to believe that $\Sigma$ does have some semantic meaning. While our efforts are not perfect, we hope our proposal for this PDC module marks a crucial step in parameterizing the uncertainty of particles' spatial states. We welcome suggestions to further conceptualize the connection between learned variance and atomic uncertainty.
>
> ------------------------------
>
> We genuinely thank your quick feedback and value the opportunity to discuss this with you. If you have any further suggestions or specific areas you would like us to focus on during revision, please feel free to let us know.

---

### Official Review · Reviewer_xfqG · 2023-10-31

**Soundness:** 2 fair
**Presentation:** 3 good
**Contribution:** 3 good
**Rating:** 6
**Confidence:** 4

**Summary:**

This paper proposes a novel deep learning architecture, Refine-PPI, for protein-protein binding mutation effect (DDG) prediction. Refine-PPI consists of two modules. The first module learns to predict the mutated structure through a masked mutation modeling task on wild-type structures. The second module learns to predict DDG of a protein-protein complex based on wild-type and mutated structures. The second module represents a protein-protein complex as a probabilistic density cloud (PDC) and encodes it using a novel PDC-GNN, where the messages are represented by its mean and variance. Refine-PPI achieves state-of-the-art performance on the standard SKEMPI benchmark.

**Strengths:**

* This paper models a protein as a dynamic structure, using a probabilistic density cloud representation.
* This paper develops a new message passing network architecture for probabilistic density clouds. The messages between each node consists of both mean and variance.
* The evaluation setup is comprehensive, with all the relevant baselines

**Weaknesses:**

* The model only slightly outperforms previous state-of-the-art RDE-Net on a subset of metrics. It seems that overall performance of Refine-PPI and RDE-Net is similar.
* The evaluation of Refine-PPI on the first mutation structure prediction task is missing.

**Questions:**

* The description of probabilistic density cloud representation is a bit unclear. In particular, how are the $sigma$'s initialized? If they are initialized as zero, then they will stay zero all the time. Are they initialized by some physical calculations?
* For the first task of mutation structure prediction task, can you report the RMSD between predicted mutated structure and ground truth?

---

> ### Author Response · Authors · 2023-11-11
> **Reponse to Reviewer xfqG**
>
> We sincerely appreciate your time and effort in reviewing our paper, and we are delighted that you identified the strengths of our approach, particularly the dynamic modeling of proteins and the development of a novel message-passing network architecture for probabilistic density clouds. Your insightful feedback has been invaluable, and we would like to address the points raised in your review as follows.
>
> (1) We appreciate your honesty in highlighting the weaknesses of our work and understand your concern regarding the marginal improvement over the previous state-of-the-art model, RDE-Net. However, it is worth noting that RDE is pretrained on the additional protein structure dataset, PDB-REDO, which contains more than 130K refined X-ray structures in Protein Data Bank. Meanwhile, our Refine-PPI adopts no pretraining strategy and is directly trained on Skempi v2 with only 345 wide-type structures. Refine-PPI enjoys no benefits of unsupervised pretraining but achieves competitive or even better performance than existing algorithms, underscoring the superiority of our architecture design. As mentioned in Appendix C, several prior studies [A, B, C] have demonstrated that structural pretraining is beneficial to dramatically expand the representation space of deep learning models, and it is promising to pretrain Refine-PPI with more experimental protein structures and transfer the knowledge to this mutation effect prediction task. If we leverage the same PDB-REDO for pretraining, the per-structure Spearman correlation will significantly increase from 0.41 to 0.44 (see Table below for a clear comparison).
>
> Moreover, as declared in both our paper and RDE-Net, the correlation for one specific protein complex is often of greater interest and importance in practical applications. It is preferred to attach more attention to the average per-structure Pearson and Spearman correlation coefficients rather than the overall metrics.
>
> |   Method   | Pretrain | Per-Structure | Per-Structure |   Overall  |   Overall  |   Overall  |   Overall  |   Overall  |
> |:----------:|:--------:|:-------------:|:-------------:|:----------:|:----------:|:----------:|:----------:|:----------:|
> |            |          |    Pearson    |    Spearman   |   Pearson  |  Spearman  |    RMSE    |     MAE    |    AUROC   |
> |   RDE-Net  |    Yes   |     0.4448    |     0.4010    |   0.6447   |   0.5584   |   0.5799   |   1.1123   |   0.7454   |
> | Refine-PPI |    No    |     0.4475    |     0.4102    |   0.6584   |   0.5394   | **1.5556** | **1.0946** |   0.7517   |
> | Refine-PPI |    Yes   |   **0.4561**  |   **0.4374**  | **0.6592** | **0.5608** |   1.5643   |   1.1093   | **0.7542** |
>
>
> [A] Zhang, Zuobai, et al. "Protein representation learning by geometric structure pretraining." arXiv preprint arXiv:2203.06125 (2022).
>
> [B] Chen, Can, et al. "Structure-aware protein self-supervised learning." Bioinformatics 39.4 (2023): btad189.
>
> [C] Wu, Fang, et al. "Pre‐Training of Equivariant Graph Matching Networks with Conformation Flexibility for Drug Binding." Advanced Science 9.33 (2022): 2203796.
>
> (2) For the first task of mutation structure prediction, we apologize for not reporting the RMSD between the predicted mutated structure and the ground truth. This information is, indeed, very crucial. However, it is worth mentioning that obtaining the structures of mutant proteins is a persistent challenge, as they are often elusive to acquire. All mutant structures in Skempi V2 are not available and we only have 345 wide-type structures. Therefore, we are afraid that we are unable to report the exact structure prediction error between the predicted mutated structure and ground truth. Notably, the inaccessibility of mutant structures is the core motivation for our structure refinement module, which is first trained by a mask mutation modeling (MMM) task on available wide-type structures and then transferred to hallucinate the inaccessible mutant protein structures. During the MMM training phase, the RMSD between predicted wide-type structures and ground truth is 1.437, which will increase if we enlarge the length of the masked region. We hope this statistic can provide insight into the modeling accuracy and partially answer your question.

---

> ### Author Response · Authors · 2023-11-11
> **Reponse to Reviewer xfqG (Part II)**
>
> (3) Regarding the initialization of probabilistic density cloud representation, we agree that our description needs clarification. Specifically, we investigate three sorts of initialization mechanisms for $\sigma$. First and naively, we turn all $\sigma$ to be equivalent to one (a unit length). Second, we depend on physical principles and utilize molecular dynamic (MD) simulations to attain the short motion trajectories (10 nano-seconds) of these complexes in the 3D space. Then we calculate the root-mean-square fluctuation (RMSF) of each amino acid and take this value as the initial input of $\sigma$. Third, we adopt a learnable strategy to initialize $\sigma$. To be explicit, an embedding layer is created for each category of 20 residue types to a 3-dimensional continuous vector. This routine learns the variance of different components completely from the data.
>
> The performance of different initialization approaches is listed below, and it can be found that constant initialization is the worst. Besides, the MD-based methodology outperforms slightly better results than the embedding-based one. However, since MD simulations are time-consuming and costly, it is prohibited to implement MD during the inference stage each time. As a consequence, we use the third sort in our paper. In the revised manuscript, we will provide a more detailed explanation of the initialization process (e.g., in the Appendix).
>
> |       Method       | Per-Structure | Per-Structure |
> |:------------------:|:-------------:|:-------------:|
> |                    |    Pearson    |    Spearman   |
> |     Unit Length    |     0.4422    |     0.4043    |
> |   MD Simulations   |    **0.4522**    |   **0.4287**    |
> | Learnable Variance |     0.4475    |     0.4102    |
> --------------------
> Once again, we appreciate your thorough review and constructive feedback. Your insights will undoubtedly contribute to the refinement and improvement of our work. We will submit a revised version that addresses the mentioned concerns and provides a clearer understanding of our contributions.

---

> ### Comment · Reviewer_xfqG · 2023-11-21
> **Thank you for your response**
>
> Based on other reviewer's comments, I think it's important to report the standard deviation of your method (e.g. 5-fold cross validation) and see how significant your improvement is. Also, it would be good to compare your side-chain reconstruction (MMM task) with existing methods (e.g., FoldX or RDENet side-chain packing) on the wildtypes. I will keep my original score nevertheless.

---

### Official Review · Reviewer_cuXz · 2023-11-03

**Soundness:** 1 poor
**Presentation:** 3 good
**Contribution:** 2 fair
**Rating:** 3
**Confidence:** 4

**Summary:**

A framework, Refine-PPI, of predicting $\Delta\Delta G$ is proposed in this paper, which include 3 components: a structure encoder $h_\rho$, a structure refiner $f_\theta$, and a readout (pooling) function $g_\tau$.

New backbone coined as PDC-Net is proposed to model structures.

Experimental results show marginal improvements on $\Delta\Delta G$ prediction.

**Strengths:**

The general framework of Refine-PPI is interesting.

The $\Delta\Delta G$ prediction quality is seemingly equivalent / marginally improved.

**Weaknesses:**

1. A lot of missing details hinder the reproducibility of the paper. See Questions.

2. Overall the paper introduce a new framework and a new architecture. Although a general ablation is done, the paper very much lacks deep analysis to each components.

3. Weak benchmark performance: the elevation of performance is too marginal (especially those in the Appendix), and the time complexity is not studied. No variance is reported.

4. The usage of term "thermodynamics" and "hallucination" is hardly relevant to the proposed method and thus confusing. I would suggest the authors to use plainer descriptions.

**Questions:**

Q1 How are $h_\rho, f_\theta, g_\tau$ built precisely? What are the $\phi$ functions in Eq 5-7?

Q2 How are $\sigma_{x_i}$s modeled? And how are those initialized? In Eq 4 they are matrices while in Eq 7 they seem to be vectors. In my opinion 3d variance should either be a scalar or a learnable SPD matrix. Using vectors does not satisfy SE(3) invariance because an ellipsoid with standard axis is presumed, and the results are thus varied when rotations are applied to the input structure. Thus the method is not "readily apparent" to be equivalent.

Q3 There lacks a visualization of the "hallucinated" structures.

Q4 On what data, precisely, is Refine-PPI trained? The description seems to point to a 3-fold cross validation, but the concrete splits should be specified. And, since all benchmark performances are "directly copied" from a preprint, the authors must justify that their evaluation scheme is exactly the same to all benchmarks.

---

> ### Author Response · Authors · 2023-11-12
> **Response to Reviewer cuXz**
>
> Dear Reviewer cuXz, we sincerely appreciate your comprehensive comment on our paper. Your thoughtful comments are invaluable, and we are committed to addressing the concerns raised in your review. Below, we provide detailed responses to each point:
>
> (1) First and foremost, we apologize for any confusion. To elucidate, the backbone module $h_\rho$ can take the form of any conventional geometric neural networks (e.g., GVP-GNN, EGNN, SE(3)-Transformer, Graph-Transformer). Here, we adopt a one-layer Graph Transformer [A,B] to extract general representations of proteins. The refinement model $f_\theta$ needs to output both updated features and coordinates, and, therefore, we use PDC-EGNN as $f_\theta$ in our experiments. Lastly, the head predictor $g_\tau$ is a simple linear layer that accepts the concatenation of both wide-type and mutation-type representations and forecasts the free energy change. The total model size of Refine-PPI is approximately 6M. In addition, $\phi_e, \phi_h, \phi_{\mu}, \phi_\sigma$ are the edge, node, mean, and variance operations respectively that are commonly approximated by Multilayer Perceptrons (MLPs). We will provide explicit details on the construction of these notations to enhance understanding in our revised paper. Thanks for your question to improve the clarity.
>
> [A] Shan, Sisi, et al. "Deep learning guided optimization of human antibody against SARS-CoV-2 variants with broad neutralization." Proceedings of the National Academy of Sciences 119.11 (2022): e2122954119.
>
> [B] Luo, Shitong, et al. "Antigen-specific antibody design and optimization with diffusion-based generative models for protein structures." Advances in Neural Information Processing Systems 35 (2022): 9754-9767.
>
> (2) We recognize your concern about the usage of terms like "thermodynamics" and "hallucination." Please allow us to explain our motivations for using these words.
>
> i) Thermodynamics [A] is a branch of physics that deals with heat, work, and temperature, and their relation to energy, entropy, and the physical properties of matter and radiation. It applies to a wide variety of topics in science and engineering, especially physical chemistry, and biochemistry. In our scenario, biologically relevant macromolecules, such as proteins, do not operate as static, isolated entities. On the contrary, they are involved in numerous interactions with other species, such as proteins, nucleic acid, membranes, small molecule ligands, and also, critically, solvent molecules. Like any other spontaneous process, binding occurs only when it is associated with a negative Gibbs' free energy of binding ($\Delta G$), which may have differing thermodynamic signatures, varying from enthalpy- to entropy-driven. Thus, the understanding of the forces driving the recognition and interaction require a detailed description of the binding thermodynamics, and a correlation of the thermodynamic parameters with the structures of interacting partners. Such an understanding of the nature of the recognition phenomena is of a great importance for medicinal chemistry research, since it enables truly rational structure-based molecular design. A great number of studies have explored the contribution of protein dynamics to the binding thermodynamics and kenetics of drug-like compounds [B, C, D, E, F]. Here, we adopt this term to express the need for accurately modeling biomoecules' inherent dynamics.
>
> [A] Wikipedia. https://en.wikipedia.org/wiki/Thermodynamics
>
> [B] K., A. (2011). Thermodynamics of Ligand-Protein Interactions: Implications for Molecular Design. InTech. doi: 10.5772/19447
>
> [C] Khatri, K.S., Modi, P., Sharma, S., Deep, S. (2020). Thermodynamics of Protein-Ligand Binding. In: Singh, D., Tripathi, T. (eds) Frontiers in Protein Structure, Function, and Dynamics. Springer, Singapore. https://doi.org/10.1007/978-981-15-5530-5_7
>
> [D] Olsson, Tjelvar SG, et al. "The thermodynamics of protein–ligand interaction and solvation: insights for ligand design." Journal of molecular biology 384.4 (2008): 1002-1017.
>
> [E] Amaral, Marta, et al. "Protein conformational flexibility modulates kinetics and thermodynamics of drug binding." Nature communications 8.1 (2017): 2276.
>
> [F] Zheng, Li-E., et al. "Machine Learning Generation of Dynamic Protein Conformational Ensembles." Molecules 28.10 (2023): 4047.

---

> ### Author Response · Authors · 2023-11-12
> **Response to Reviewer cuXz (Part II)**
>
> ii) Previous studies exemplified by Google’s DeepDream train networks to recognize faces and other patterns in images, and invert and adjust arbitrary input images to draw more strongly resemble faces or other patterns as perceived by the network. The generated images are often referred to as **hallucinations** because they may not faithfully represent any actual face, but what DL models view as an ideal face. Remarkably, this mechanism has also demonstrated success in the context of macromolecules. It has been shown that the information stored in the many parameters of the trained networks can be harnessed to design new protein structures featuring new sequences. A lot of recent works [A, B, C, D] published in top journals have used deep network hallucination to generate a wide range of functional proteins. In our algorithm, we take a similar methodology and explore whether networks trained on existing wide-type structures could be inverted to generate brand new “ideal” protein structures according to the mutant information.
>
> [A] Anishchenko, ... & Baker, D. (2021). De novo protein design by deep network hallucination. Nature, 600(7889), 547-552.
>
> [B] Wicky, ... & Baker, D. (2022). Hallucinating symmetric protein assemblies. Science, 378(6615), 56-61.
>
> [C] An, Linna, et al. "Hallucination of closed repeat proteins containing central pockets." Nature Structural & Molecular Biology (2023): 1-6.
>
> [D] Costello, Zak, and Hector Garcia Martin. "How to hallucinate functional proteins." arXiv preprint arXiv:1903.00458 (2019).
>
> -------------------------------
> However, while we find these terms conceptually appropriate, we fully understand that these words can be confusing for some readers. We would like to replace them with more precise and relevant descriptors for improved clarity. Do you have any suggestions or recommendations for the title? We are very pleased to modify it with your opinion.
>
>
> (3) We appreciate your honesty in highlighting the weaknesses of our work and understand your concern regarding the marginal improvement over the previous state-of-the-art model, which has also been mentioned by Reviewer xfqG. However, it is worth noting that RDE is pretrained on the additional protein structure dataset, PDB-REDO, which contains more than 130K refined X-ray structures in Protein Data Bank. Meanwhile, our Refine-PPI adopts no pretraining strategy and is directly trained on Skempi v2 with only 345 wide-type structures. Refine-PPI enjoys no benefits of unsupervised pretraining but achieves competitive or even better performance than existing algorithms, underscoring the superiority of our architecture design. As mentioned in Appendix C, several prior studies [A, B, C] have demonstrated that structural pretraining is beneficial to dramatically expand the representation space of deep learning models, and it is promising to pretrain Refine-PPI with more experimental protein structures and transfer the knowledge to this mutation effect prediction task. If we leverage the same PDB-REDO for pretraining, **the per-structure Spearman correlation will significantly increase from 0.41 to 0.44** (see Table below for clear comparison). Notably, We emphasize the importance of per-structure metrics over overall metrics for practical applications.
>
> |   Method   | Pretrain | Per-Structure | Per-Structure |   Overall  |   Overall  |   Overall  |   Overall  |   Overall  |
> |:----------:|:--------:|:-------------:|:-------------:|:----------:|:----------:|:----------:|:----------:|:----------:|
> |            |          |    Pearson    |    Spearman   |   Pearson  |  Spearman  |    RMSE    |     MAE    |    AUROC   |
> |   RDE-Net  |    Yes   |     0.4448    |     0.4010    |   0.6447   |   0.5584   |   0.5799   |   1.1123   |   0.7454   |
> | Refine-PPI |    No    |     0.4475    |     0.4102    |   0.6584   |   0.5394   | **1.5556** | **1.0946** |   0.7517   |
> | Refine-PPI |    Yes   |   **0.4561**  |   **0.4374**  | **0.6592** | **0.5608** |   1.5643   |   1.1093   | **0.7542** |
>
> Besides, as for the time complexity of Refine-PPI, we empirically verify that it has little difference from traditional deep learning-based methodologies. To be explicit, it spends RDE and Refine-PPI about 60 and 66 seconds, separately, to perform 3-fold inference predictions of all 6.7K entries in Skempi V2. The generation process of mutant structures is very fast, making it feasible for real-world applications
>
> |   Method   | Inference Time |
> |:----------:|:--------:|
> |   RDE-Net  | 60 s |
> | Refine-PPI | 66 s |
>
> [A] Zhang, Zuobai, et al. "Protein representation learning by geometric structure pretraining." ICLR 2023.
>
> [B] Chen, Can, et al. "Structure-aware protein self-supervised learning." Bioinformatics 39.4 (2023): btad189.
>
> [C] Wu, Fang, et al. "Pre‐Training of Equivariant Graph Matching Networks with Conformation Flexibility for Drug Binding." Advanced Science 9.33 (2022): 2203796.

---

> ### Author Response · Authors · 2023-11-12
> **Response to Reviewer cuXz (Part III)**
>
> (4)  We appreciate your clarification on the use of $\sigma$ and how to initialize them in our probability density cloud representation.
> i) You are correct that $\sigma$ are matrices in Equ. 4 while are vectors in Equ. 7. The main reason is that $\boldsymbol{\Sigma}_i\in \mathbb{R}^{3\times 3}$ is a diagonal covariance matrix indicating that different axes are independent of each other. Consequently, we utilize $\boldsymbol{\sigma}$ to denote the diagonal elements of the variance matrix $\mathbf{\Sigma}$ for simplicity.  we will incorporate this into our revised manuscript.
>
> ii) Specifically, we investigate three sorts of initialization mechanisms for $\sigma$. First and naively, we turn all $\sigma$ to be equivalent to one (a unit length). Second, we depend on physical principles and utilize molecular dynamic (MD) simulations to attain the short motion trajectories (10 nano-seconds) of these complexes in the 3D space. Then we calculate the root-mean-square fluctuation (RMSF) of each amino acid and take this value as the initial input of $\sigma$. Third, we adopt a learnable strategy to initialize $\sigma$. To be explicit, an embedding layer is created for each category of 20 residue types to a 3-dimensional continuous vector. This routine learns the variance of different components completely from the data.
>
> The performance of different initialization approaches is listed below, and it can be found that constant initialization is the worst. Besides, the MD-based methodology outperforms slightly better results than the embedding-based one. However, since MD simulations are time-consuming and costly, it is prohibited to implement MD during the inference stage each time. As a consequence, we use the third sort in our paper. In the revised manuscript, we will provide a more detailed explanation of the initialization process (e.g., in the Appendix).
>
> --------------------
> |       Method       | Per-Structure | Per-Structure |
> |:------------------:|:-------------:|:-------------:|
> |                    |    Pearson    |    Spearman   |
> |     Unit Length    |     0.4422    |     0.4043    |
> |   MD Simulations   |     0.4522    |     0.4287    |
> | Learnable Variance |     0.4475    |     0.4102    |
>
> (5) We acknowledge the absence of visualization for "hallucinated" structures. In the revision, we will include visualizations (see Appendix) to provide a clearer understanding of the proposed method.
>
> (6) We apologize for any confusion regarding the training data and evaluation scheme. In our evaluation, we adopt the same 3-fold cross-validation split as RDE-Net. To be specific, the numbers of training and validation samples for fold-0, fold-1, and fold-2 are 4777/1929, 4290/2416, and 4345/2361. We confirm that our splitting strategy completely accords with all benchmarks reported in the RDE paper. Besides that, we would like to clarify that RDE was officially accepted by last year's ICLR [A] and it is our fault to use its preprint version as the reference.
>
> [A] Rotamer Density Estimator is an Unsupervised Learner of the Effect of Mutations on Protein-Protein Interaction. ICLR 2023. https://openreview.net/forum?id=_X9Yl1K2mD
>
> ---------------------------------------------------
> We would like to extend our sincere appreciation for the meticulous review you have conducted and the valuable insights you have shared. Your constructive feedback is highly valuable to us, and we are fully committed to preparing a revised version that meticulously addresses all the raised concerns. We aim to enhance the clarity of our contributions and ensure that the revised manuscript aligns seamlessly with your feedback.

---

> > ### Comment · Reviewer_cuXz · 2023-11-14
> > **reply to rebuttal**
> >
> > Thanks for the detailed rebuttal contents. It seems that implementation details are enlarged (I think since we are in ICLR the authors are free to revise their paper, besides adding the rebuttal contents), and the authors did some experiments, but my concerns are not addressed.
> >
> > 1) the shown improvements, even with pretrain, are still marginal. (seems that the RMSE of baseline is buggy)
> >
> > 2) in iii) the authors argues that as long as the covariance matrix is SPD the model is equivariant, but this is wrong. Using diagonal matrices presumes that the variance are independent along standard axis. This is not SO(3) equivariant.
> >
> > 3) Visualization is discussed but I can't find any.

---

> ### Author Response · Authors · 2023-11-14
> **Response to Reply**
>
> Thank you for your thoughtful engagement with our work. We appreciate the time and effort you invested in examining our response to your concerns so quickly. We are glad to know that you recognize the significant enlargement of implementation details. Please see our latest response to your reply below:
>
> (1) We understand your concern that the improvement of Refine-PPI in RMSE is marginal over baselines. However, as pointed out by RDE [A] and DiffAffinity [B], **these per-structure correlation scores are more relevant to practical applications and should better reflect how good the models are for real-world challenges**. Biologists and protein designers typically prioritize whether binding affinity magnitude improves after mutations, rather than the absolute value of $\Delta\Delta G$. More importantly, **the scalar values of experimental $\Delta \Delta G$ in Skempi v2 are not standardized** [B], and the robustness of $\Delta \Delta G$ is affected by factors such as batch effects and environmental fluctuations. Consequently, ranking coefficients such as Spearman and Pearson's correlations are always regarded as much more vital metrics in evaluating the effectiveness of different algorithms in predicting binding affinity or protein stability [B, C, D]. Our model selection during training is based on Spearman's correlation on the validation set, prioritizing it over RMSD or MAE.
>
> [A] Rotamer Density Estimator is an Unsupervised Learner of the Effect of Mutations on Protein-Protein Interaction. ICLR 2023.
>
> [B] Liu, Shiwei, et al. "Predicting mutational effects on protein-protein binding via a side-chain diffusion probabilistic model." NIPS 2023.
>
> [C] Öztürk, Hakime, Arzucan Özgür, and Elif Ozkirimli. "DeepDTA: deep drug–target binding affinity prediction." Bioinformatics 34.17 (2018): i821-i829.
>
> [D] Kim, Ryangguk, and Jeffrey Skolnick. "Assessment of programs for ligand binding affinity prediction." Journal of computational chemistry 29.8 (2008): 1316-1331.
>
> (2) We are grateful for your clarification regarding the equivariance property. Upon reevaluation, we have determined that our initial assumption of the covariance matrix being a diagonal matrix is insufficient to guarantee equivariance.  As a solution, we take a further step and posit that **the covariance matrix is an isotropic matrix**, meaning that all elements on the diagonal are equal.  Under this condition, our Refine-PPI ensures that for any rotation $Q\in \mathbb{R}^{3\times 3}$ and any translation $g \in \mathbb{R}^{3}$, we have
>
> $\mathbf{h}^{(l+1)}, \{Q \boldsymbol{\mu} ^{(l+1)} + g, Q^\top \boldsymbol{\Sigma}^{(l+1)}Q \}= \textrm{PDC-L}[\mathbf{h}^{(l)}, \{Q \boldsymbol{\mu} ^{(l)} + g, Q^\top\boldsymbol{\Sigma} ^{(l)}Q \}, \mathcal{E}].$
>
> There, $\boldsymbol{\Sigma}$ can be initialized by a pre-defined value (e.g., an identity matrix or RMSF in MD simulations) or a learnable embedding. We further assessed the performance difference between this new covariance matrix setting (i.e., isotropic matrix) and the previous one (diagonal matrix), with the results summarized below. Notably, there is no significant difference observed using the spherical matrix as the covariance. To elucidate this, we examined the previously learned diagonal matrix and discovered that all elements in the diagonal are very close to each other. In essence, even without imposing the isotropic restriction, the model inherently learned to assign similar values to diagonal elements. This perspective sheds light on the necessity of isotropy in our covariance matrix.
>
> |       Method       | Per-Structure | Per-Structure |
> |:------------------:|:-------------:|:-------------:|
> |                    |    Pearson    |    Spearman   |
> |     Isotropic Matrix   |     0.4461    |     **0.4106**    |
> |    Diagonal Matrix  |    **0.4475**    |   0.4102    |
>
> We have provided a detailed proof of the equivariance of Refine-PPI in Appendix D and would value your review to enhance our theoretical foundations.
>
> (3) We apologize for any confusion regarding the visualization aspect. We have already revised our paper once on 13th November, but perhaps the updated version is delayed from being attained by the reviewers because of some unknown systematic reasons in Openreview. Just confirm that we have uploaded the new manuscript again, and please see Appendix C for the visualization of hallucinated structure examples. If you encounter difficulties accessing our updated manuscript, kindly inform us. Your keen eye for such details is instrumental in enhancing the overall quality of our work.
>
> ------------------------------------------
>
> Once again, we appreciate your dedication to the peer-review process, and we are committed to addressing each of your concerns to improve the clarity, correctness, and overall contribution of our paper. Thank you for your constructive feedback.

---

### Author Response · Authors · 2023-11-13
**Official Comment by Authors**

To begin with, we would like to express our sincere gratitude to all three reviewers and AC for dedicating time and effort to reviewing our manuscript. Your constructive feedback has been invaluable in refining our work. Below is a summary of our responses to some comments that have been raised by more than one reviewer:

(1) Firstly, we acknowledge all reviewers' concerns regarding the marginal improvement over the previous state-of-the-art model. It's crucial to note that many robust baselines, including RDE-Net and MIF-Net, achieve high performance through pretraining on the extensive PDB-REDO dataset, comprising over 130,000 refined X-ray structures in the Protein Data Bank. In contrast, our Refine-PPI opts for no pretraining strategy, directly training on Skempi v2 with only 345 wild-type structures. Despite lacking the advantages of unsupervised pretraining, Refine-PPI demonstrates competitive or superior performance compared to existing algorithms, highlighting the efficacy of our architectural design.

Earlier research studies have established the benefits of structural pretraining in significantly expanding the representation space of deep learning models. It holds promise to pretrain Refine-PPI with a broader set of experimental protein structures, transferring the acquired knowledge to the mutation effect prediction task. In response to Reviewer Criu's suggestion and to substantiate our claim, we conducted pretraining using the same PDB-REDO dataset. Remarkably, the per-structure Spearman correlation shows a substantial increase from 0.41 to 0.44 (refer to the table below). This outcome underscores the significance of per-structure metrics, particularly in practical applications.

|   Method   | Pretrain | Per-Structure | Per-Structure |   Overall  |   Overall  |   Overall  |   Overall  |   Overall  |
|:----------:|:--------:|:-------------:|:-------------:|:----------:|:----------:|:----------:|:----------:|:----------:|
|            |          |    Pearson    |    Spearman   |   Pearson  |  Spearman  |    RMSE    |     MAE    |    AUROC   |
|   RDE-Net  |    Yes   |     0.4448    |     0.4010    |   0.6447   |   0.5584   |   0.5799   |   1.1123   |   0.7454   |
| Refine-PPI |    No    |     0.4475    |     0.4102    |   0.6584   |   0.5394   | **1.5556** | **1.0946** |   0.7517   |
| Refine-PPI |    Yes   |   **0.4561**  |   **0.4374**  | **0.6592** | **0.5608** |   1.5643   |   1.1093   | **0.7542** |

(2) Secondly, we appreciate all reviewers' comments to help improve the clarity of our manuscript. For instance, all of you mention the initialization approach of coordinate variance, which is a key factor in our PDC-module but we do not provide adequate details. To be specific, we investigate three sorts of initialization mechanisms for $\sigma$: a unit length, the root-mean-square fluctuation (RMSF) obtained by MD simulations, and learnable variance. Results show that MD-based initialization outpasses others, but we opt for the learnable approach due to MD's time-consuming nature.

Besides, Reviewer cuXz and Criu both ask for exposing more experimental details, including the choice of $h_\rho$, $f_\theta$, and $g_\tau$, the split of Skempi dataset, the implementation of baseline algorithms, and ablation studies. We enrich the content of our paper according to your advice, which dramatically promotes readability and clarity.

Moreover, several reviewers have discussed the suitability of essential terms like 'thermodynamics' and 'hallucination'. We understand that our usage of these words might over-claim the contributions or be misleading for people who are not familiar with this research area. As a remedy, we plan to rename the PDC-module as 'Uncertainty-aware Neural Network' or 'Uncertainty-inspired Neural Network', with the final decision to be made after further discussion with you during this phase.

(3) Finally, we express gratitude for the meticulous examination of the mathematical underpinnings of our uncertainty module. Specifically, Reviewer cuXz contends that the variance should be a scalar or a trainable SPD matrix. Prompted by this observation, we reevaluate our prior assumption that the covariance matrix is diagonal, recognizing its inadequacy in ensuring equivariance. After careful consideration, we propose that **the covariance matrix is isotropic, meaning all diagonal elements are identical**. With this adjustment, our Refine-PPI adheres rigorously to the equivariance property, as detailed in the proof presented in Appendix D.

Meanwhile, Reviewer Criu proposes a more effective formula to update the positional variance, which uses the same $\phi_\mu$ and distributing and squaring the $\mathbf{x}_i$ terms. And we empirically discover a slight improvement in the mutant effect prediction task. We firmly believe our design of this novel architecture will pave the road to model atomic uncertainty in the molecular system.

---

> ### Author Response · Authors · 2023-11-15
> **Official Comment by Authors (Part II)**
>
> -----------------------------
> Should you have further questions or suggestions, please feel free to reach out. We hope our responses and updates are useful and constructive, and we kindly request a reconsideration of the manuscript's scores!

---

### Author Response · Authors · 2023-11-20
**A Kind Reminder for Reviewers' Feedback**

Dear reviewers,

We greatly appreciate your prevous valuable comments. In our response, we conduct additional experiments to pretrain our Refine-PPI on unlabeled PDB and achieved much higher per-structure correlations (**$0.41 \rightarrow 0.44$**), to investigate different approaches to initialize variance, and to explore a new update Equation. Furthermore, we have presented supplementary evidence highlighting the correlation between learned variance and atomic uncertainty, demonstrating that our PDC-module effectively encodes and parameterizes dynamic information implicitly. In addition, we consolidated the mathematical foundations by requiring the variance matrix to be isotropic, which garantees our network to be strictly equivariant. Additionally, we have addressed any ambiguities in our paper and incorporated all modifications into the revised manuscript for your thoughtful consideration. We trust that these adjustments address your concerns.

As you may know, unlike previous years, the discussion period this year can only last until November 22, and we are gradually approaching this deadline. We would like to discuss this with you during this time and would be happy to provide more information based on your feedback or further questions. We understand the demands on your time, especially if you are also engaged in the rebuttal process for your own submissions. However, we would greatly appreciate it if you could spare some time to review our response. Your insights are invaluable to us.

Should you find our response satisfactory, we kindly request you to consider updating your score. If you require any clarifications or have additional queries, please do not hesitate to reach out to us. Thanks!

---

### Meta-Review · Area_Chair_qLW3 · 2023-12-11

**Metareview:**

The paper presents Refine-PPI, a framework for predicting protein-protein interactions. It contains three main elements: a structure encoder, a structure refiner, and a readout function. The paper also introduces a new architecture named PDC-Net.

Reviewers expressed concerns about the limited improvement over the previous leading model, the absence of evaluation for the mutation structure prediction task, the need for more detailed implementation information, a more in-depth analysis of each component, improved benchmark performance, clearer terminology usage, and the mathematical basis of the PDC module.

Despite some issues being adequately addressed during the discussion period, reviewers retained overall negative ratings. They maintained concerns about the marginal improvement in results, the conceptual presentation of the PDC module, the claim that RMSE serves as a distributional modeling objective, and the supposed semantic correlation between the learned "variance" and positional uncertainty.

**Justification For Why Not Higher Score:**

Despite some issues being adequately addressed during the discussion period, reviewers retained overall negative ratings. They maintained concerns about the marginal improvement in results, the conceptual presentation of the PDC module, the claim that RMSE serves as a distributional modeling objective, and the supposed semantic correlation between the learned "variance" and positional uncertainty.

**Justification For Why Not Lower Score:**

N/A

---

### Decision · Program_Chairs · 2024-01-16

Reject